# Entropy-Regularized Diffusion-Policies in Offline Reinforcement Learning for Antibody Sequence Design

*Note for reviewers: revised and newly added text in response to the TMLR review round is typeset in blue. Newly added figures and tables are introduced and referenced from the blue text.*

## Abstract

The discovery of therapeutic antibodies is traditionally performed through wet lab screening, which is costly and time-consuming. Generative models offer a data-driven alternative, however such methods become unreliable outside the training distribution. We present Sequential Diffusion + Q-Learning (SeqDiff+QL), which formulates antibody sequence design as a constrained offline Reinforcement Learning (RL) problem, enforcing proximity to the training distribution. SeqDiff+QL employs an entropy-regularized diffusion policy that, through policy improvement, is trained to sequentially generate Complementarity Determining Region (CDR) sequences with higher predicted binding affinity based on a variety of training distributions. Our novel entropy regularization thereby promotes diverse candidate generation, while the integration of biophysical priors through contrastive Variational Autoencoder (VAE) latent representations improves the stability of the generative process. The framework can learn from heterogeneous sequence sources across different training distributions. Using the Absolut! simulator and Rosetta energy function as in silico affinity evaluation oracles, we show that SeqDiff+QL produces candidate sequences with improved oracle-predicted affinity across two distinct antigen targets per oracle. We further document an explicit affinity-diversity tradeoff: Q-guided variants reduce diversity and novelty relative to unguided generators, and our entropy regularizer partially recovers diversity at high affinity.

## 1 Introduction

Antibodies are a class of proteins with great potential for treating diseases such as cancer (Kaplon et al., 2023; Norman et al., 2020; Robert et al., 2022). However, the discovery of therapeutic antibodies in classical wet lab experiments is constrained by high costs and low throughput (Angermueller et al., 2019; Shanehsazzadeh et al., 2024; Vamathevan et al., 2019). Computational approaches to antibody design, such as generative models, have emerged as a powerful tool for addressing these challenges, offering a data-driven approach, significantly reducing the time and resources required (Shanehsazzadeh et al., 2024). In prior works, specific requirements in the field of protein design have been identified, translating to *algorithmic requirements* for computational approaches to protein design.

**Objective-driven design:** The ability to optimize desired properties such as target specificity and binding affinity instead of generating outputs similar to training data (Gruver et al., 2024; Shanehsazzadeh et al., 2024; Vamathevan et al., 2019). **Diversity**: Approaches should employ diverse, non-deterministic design schemes to account for the diverse nature of real-world antibody-antigen interaction (Jain et al., 2022), non-deterministic behavior of proteins (Wang et al., 2025), and the limited ability of in silico simulations to mimic real-world interaction, and overall low success rates in real-world evaluation (Vamathevan et al., 2019). **Offline Learning:** Applicability in the offline setting, which enables training on and improving upon pre-collected offline data without repeated online access to evaluation methods, as such interactions are infeasible due to the respective time and cost involved (Shanehsazzadeh et al., 2024).

A promising class of algorithms is diffusion or flow-matching models, which recently have received considerable attention due to their ability to model multimodal data distributions and generate diverse and high-quality data (Murphy, 2023). Their versatility makes them applicable to numerous tasks in the realm of protein design, including protein structure prediction (Anand & Achim, 2022; Wang et al., 2025), protein-protein docking (Ketata et al., 2023), as well as protein sequence and structure (co-)design (Alamdari et al., 2023; Campbell et al., 2024; Chen et al., 2024b; Jin et al., 2022; Kim et al., 2024; Luo et al., 2022; Martinkus et al., 2023; Verma et al., 2023; Watson et al., 2023; Zhou et al., 2024).

However, basic diffusion only models a given data distribution and does not optimize for a desired objective, such as binding affinity to an antigen, leading to the development of many guiding methods (Dhariwal & Nichol, 2021; Ho & Salimans, 2022; Park et al., 2025; Wang et al., 2023; Zheng et al., 2023). While some methods, such as Direct Preference Optimization (DPO) (Wallace et al., 2023) and Noise Contrastive Alignment (NCA) (Chen et al., 2024a), reweight samples within the data distribution according to a preference or reward, RL based methods, due to the maximization of a learned Q-function, can explicitly guide the model towards higher-reward regions of sequence space. However, learned affinity estimates are unreliable far from the training distribution, so unconstrained optimization may produce sequences with high predicted scores but low real-world evaluation performance. Therefore, optimization must balance improvement with proximity to the empirical data distribution (Fujimoto & Gu, 2021; Levine et al., 2020).

In this work, we approach the task of designing antigen-specific antibodies with maximized binding affinity. Specifically, we focus on antibody sequence design, in contrast to antibody structure design or sequence structure co-design. Similar to Angermüller et al. (2020) and Jain et al. (2022), we choose a stepwise approach to antibody sequence, one amino acid (AA) at a time, and formulate the task as an Markov Decision Process facilitating the use of RL methods to maximize affinity. We propose a new offline RL algorithm, which improves predicted affinity through policy improvement while constraining the policy to the proximity of the data distribution and maintaining diversity through entropy regularization, thereby satisfying the algorithmic requirements stated above. We evaluate our method using the Absolut! simulator (Robert et al., 2022) and the Rosetta energy function (Alford et al., 2017; Simons et al., 1997) and show:

- Continuous diffusion policies employing RL enable constrained policy improvement for antibody sequence design under a data-distribution constraint

- A entropy regularization partially mitigates the diversity loss induced by Q-guidance, yielding Pareto-improved configurations that are both more diverse and higher-affinity than non-regularized counterparts in the high-affinity regime, although the very highest predicted affinities still come at a diversity cost

- Incorporating biophysical priors into the generative process using contrastive latent representations can be beneficial if priors align with evaluation properties

- That learned Q-functions can be used to identify high-affinity sequences after the generative process

- The method supports offline learning from heterogeneous sequence distributions, including random sequences, sequences generated by methods found in literature (Dauparas et al., 2022; Vogt et al., 2023; Watson et al., 2023), and murine antibody sequences

All affinity claims in this paper refer to scores produced by the Absolut! and Rosetta in silico oracles. We do not perform wet-lab validation, and our results should not be read as evidence of real binding or therapeutic value.

## 2 Background

In this section, we provide the background on antibody sequence design, RL, and latent diffusion models.

## 2.1 Antibody Sequence Design as an RL task

Antibodies are a class of proteins, consisting of a sequence of AAs, utilized by the immune system to recognize and bind foreign molecules (antigens) with high specificity (Norman et al., 2020; Robert et al., 2022). Due to their favorable binding properties, they have become the leading class of new drugs developed (Lu et al., 2020; Norman et al., 2020). When designing an antibody sequence, each of the $L$ designed positions of an antibody sequence can be assigned one of the 20 natural AAs, resulting in a search space of $20^L$ candidates. While antibodies consist of hundreds of AAs (Chiu et al., 2019), the variable CDRs contain the majority of antigen-binding AAs (Norman et al., 2020). Furthermore, the third CDR of the heavy chain (CDRH3) has the largest influence on the antibodies' specificity (Xu & Davis, 2000). Thus, the design of CDRs and especially the CDRH3 is a frequently chosen sequence design task (Cowen-Rivers et al., 2022; Khan et al., 2022; Liu et al., 2020; Luo et al., 2022; Zhou et al., 2024). We choose to maximize the binding affinity of the antibody to the antigen as our primary objective, and measures such as novelty and diversity of generated candidates as auxiliary objectives. In this work, we evaluate two design tasks. On the one hand, we design an antibody CDRH3 region of length $L = 11$, evaluated for its binding affinity towards an antigen using the Absolut! simulator. On the other hand, we design the CDR1, CDR2, and CDR3 sequences of a single-domain antibody, with a total length of $L = 28$ AAs evaluated using the Rosetta energy function (Alford et al., 2017; Simons et al., 1997). See Figure 1(a) for a visualization of such CDR regions and the design task. The vast resulting search spaces of up to $20^{28} \approx 2.68 \cdot 10^{36}$ candidates preclude an exhaustive search, thereby underscoring the potential impact of computational antibody design in therapeutics development. We focus here on nanobody CDRH3 and CDR design with $L \in \{11, 20, 28\}$, which covers the variable regions most responsible for antigen specificity (Xu & Davis, 2000). Our method itself does not impose a length limit: the policy generates one residue at a time conditioned on the prefix, so longer designs require only a longer roll-out, more padding capacity, and potentially more training data. For nanobodies the framework regions show very limited natural sequence diversity, which is why we restrict design to the CDRs; designing entire variable domains, paired heavy-light antibodies, or longer CDRH3 loops remains an interesting direction for future work, but is out of scope here. See Section A.13 for an extended discussion.

In RL, tasks are typically formulated as MDPs. We define a deterministic MDP as a tuple $\langle S, S_0, A, P, R \rangle$, where $S$ is the set of possible states, $S_0 \subseteq S$ is the set of initial states, $A$ is the set of actions, $P(s, a) : S \times A \mapsto S$ is a deterministic transition function, and $R(s, a) : S \times A \mapsto \mathbb{R}$ is a deterministic reward function.

Like Jain et al. (2022) and Angermueller et al. (2019), we choose to frame the task of designing discrete AA sequences as a stepwise generation process where the AAs are placed in the sequence one after the other. Thus, we define the set of states $S$ as the set of all possible AA sequences up to length $L$, including the empty sequence. We define the set $S_0$ as an empty AA sequence. The set $A$ is then defined as the set of 20 natural AAs. Note that we use the symbol $a$ throughout this work to refer to the RL action and the AA it represents. Consequently, we define $P(s, a) = s \,\|\, a$ as the concatenation of the sequence generated thus far with the next AA, extending the sequence by one more AA. To prevent variable-size representations for $s$, we use padding tokens and a fixed sequence length $L$. The reward function $R(s, a)$ is defined corresponding to the predicted free energy using the Absolut! simulator or Rosetta energy function. As sequences of length shorter than $L$ AAs can not be evaluated in our tasks, the reward function is sparse, returning the predicted free energy for sequences of length $L$ and a reward of 0 for all shorter sequences.

The objective in RL is to learn a policy $\pi$ that maximizes the expected sum of rewards. The action-value function $Q$ represents this expected sum starting from a given state $s_t$. As the search space of CDR sequences is huge, we estimate $\pi$ and $Q$ with function approximations $\pi_\theta$ and $Q_\phi(s, a)$, parameterized by $\theta$ and $\phi$ respectively. We define $Q_\phi(s_t, a_t) := \mathbb{E}_{\pi_\theta}[R(s_t, a_t) + \sum_{i=1}^{L-1-t} R(s_{t+i}, a_{t+i}) | a_{t+i} \sim \pi_\theta(a_{t+i}|s_{t+i})]$. An optimal policy $\pi$ thus selects the action $a$ that maximizes $Q$ for each state $s$. In our method, the policy $\pi_\theta(a_t|s_t)$ will be implemented as a continuous diffusion policy. Thus, the action $a_t \sim \pi_\theta(a_t|s_t)$ will not be a categorical AA but instead a two-dimensional continuous representation $\boldsymbol{a}_t$ encoding the respective AA. To create such a two-dimensional representation, we train a VAE to encode the 20 AAs into a two-dimensional latent space. We freeze the VAE during the training of the policy. Keep in mind that we only use two-dimensional VAE latent representations for actions, not for states $s$.

In line with the algorithmic requirements identified above, we focus on the offline RL setting, where the agent is trained using only a pre-collected dataset, which we consider well-suited for the antibody design task, as continuous interactive access to a wet lab is not always feasible (Jain et al., 2022).

## 2.2 Diffusion Models

Diffusion Models are a class of generative models that learn to generate data by iteratively denoising samples from a Gaussian noise distribution. They employ a *forward process*, or *diffusion process*, to gradually corrupt observed data into noisy data and learn a *reverse process*, or *denoising process*, to undo the corruption. A trained model can then be used to generate high-quality data from noise (Murphy, 2023).

In this work, we are dealing with both diffusion steps $n \in \{0, ..., N\}$ and RL time steps $t \in \{0, ..., T\}$. To facilitate clarity, we will use superscripts for diffusion steps and subscripts for time steps. Diffusion probabilistic models (Ho et al., 2020; Sohl-Dickstein et al., 2015) are a class of latent variable models defined as $p_\theta(x^0) := \int p_\theta(x^{0:N})dx^{1:N}$. Here, $x^1, ..., x^N$ are latent variables of the same dimensionality as the data sample $x^0$ drawn from the observed data distribution $q(x^0)$. In our setting, these data samples are embeddings $a$ of AAs drawn from a VAE latent space. The forward process gradually adds Gaussian noise to $x^0$ according to a noise schedule $\beta^1, ..., \beta^N$, over $N$ steps (Ho et al., 2020). In particular, the forward process is defined as $q(x^{1:N}|x^0) := \prod_{n=1}^{N} q(x^n|x^{n-1})$, with a single step transition $q(x^n|x^{n-1}) := \mathcal{N}(x^n; \sqrt{1-\beta^n}x^{n-1}, \beta^n\mathbf{I})$.

The reverse process is the joint distribution $p_\theta(x^{0:N})$ defined as a Markov chain starting at $p(x^N) = \mathcal{N}(x^N; 0, \mathbf{I})$ given as $p_\theta(x^{0:N}) := p(x^N)\prod_{n=1}^{N} p_\theta(x^{n-1}|x^n)$, with a learned Gaussian transition $p_\theta(x^{n-1}|x^n)$. The objective of training $p_\theta$ is to maximize the expected log-likelihood of the data, given by the evidence lower bound (ELBO) $\mathbb{E}_q[\log \frac{p_\theta(x^{0:N})}{q(x^{1:N}|x^0)}]$. In essence, the objective is the reconstruction of a sample $x^0$ from a corresponding noisy sample $x^N$. This is achieved by training a noise model $\epsilon_\theta$ to predict the noise introduced at each diffusion step (Ho et al., 2020; Murphy, 2023). Consequently, the loss for the diffusion model given a dataset $D$ can be simplified to $L(\theta) = \mathbb{E}_{n\sim\text{Unif}(1,N),\epsilon\sim\mathcal{N}(0,\mathbf{I}),x^0\sim D}[||\epsilon - \epsilon_\theta(\sqrt{\bar{\alpha}^n}x^0 + \sqrt{1-\bar{\alpha}^n}\epsilon, n)||^2]$ where $\alpha^n := 1 - \beta^n$ and $\bar{\alpha}^n := \prod_{i=1}^{n} \alpha^i$.

## 3 Related Work

The field of antibody design has been studied at the level of amino acid sequences, 3D structures, and joint sequence–structure co-design (Alamdari et al., 2023; Chen et al., 2024b; Gruver et al., 2024; Li et al., 2023; Watson et al., 2023; Campbell et al., 2024; Jin et al., 2022; Kim et al., 2024; Luo et al., 2022; Martinkus et al., 2023; Verma et al., 2023; Zhou et al., 2024) and let to the development of antibodies with confirmed real-world applicability (Dauparas et al., 2022; Gruver et al., 2024; Li et al., 2023; Vázquez Torres et al., 2025). An overview of this broad field can be found in recent work by Tang et al. (2024). In our work, we will focus on the task of antibody sequence design. Sequence design approaches differ by their interaction regime. The online regime allows repeated interaction with evaluation feedback and is well-suited for methods such as Bayesian optimization and online RL (Cowen-Rivers et al., 2022; Khan et al., 2022; Vogt et al., 2023). Active learning approaches iteratively retrain models with newly evaluated sequences. In this setting, ensembles of evolutionary algorithms (Angermüller et al., 2020), RL algorithms (Angermueller et al., 2019), and Generative Flow Networks (GFlowNets) (Jain et al., 2022) have been employed as generative models. In contrast, offline methods train once on a fixed dataset and optimize without further interaction (Chen et al., 2024b; Gruver et al., 2024; Jain et al., 2022; Li et al., 2023). We adopt this last setting, which we estimate to be most suited given the high cost and time involved when evaluating designed antibody sequences (Shanehsazzadeh et al., 2024), and present contributions in this regime here.

In their approach, Chen et al. (2024b) utilize a continuous diffusion model to generate entire antimicrobial peptide (AMP) sequences in an ESM-2 (Lin et al., 2023) latent space. They demonstrated that generated peptides exhibited similar physicochemical properties to natural peptides and aligned closely with respect to AA diversity, which highlights the expressive power of their method. However, they do not employ any technique to facilitate objective-driven design. In contrast, Gruver et al. (2024) employ discrete diffusion, whereby sequences are directly sampled in the discrete sequence space. Such discrete diffusion models are not suited for naive gradient-based guidance, as the categorical sampling at each intermediate step

prevents gradient propagation. To facilitate guidance, Gruver et al. (2024) propose sharing some of the diffusion models' hidden layers with a learned value function. Guidance is then applied only to the shared continuous latent space of the diffusion model and value model, by utilizing the gradient of the value model for optimization (Gruver et al., 2024). Li et al. (2023) fine-tune pre-trained language models on experimental data to predict binding affinity and uncertainty. Subsequently, Bayesian optimization is used on the learned model to design large and diverse libraries of high-affinity single-chain variable fragments. In our work, we combine and apply recent advances in diffusion models and offline RL to the protein sequence design task. Diffusion models, which are capable of modeling complex multi-modal distributions (Celik et al., 2025; Park et al., 2025; Ho et al., 2020; Wang et al., 2023), appear well suited for the complex circumstances underlying AA sequence design and diverse design schemes required for real-world applicability (Jain et al., 2022; Vamathevan et al., 2019; Wang et al., 2025). In the offline RL setting, it is often necessary to constrain the learned policy to the proximity of the data distribution to prevent exploiting erroneously high estimated actions. Fujimoto & Gu (2021) showed that combining a behavior cloning (BC) loss term and policy improvement is an effective approach in many offline RL domains. In their work, Wang et al. (2023) extend this core idea to diffusion policies, alleviating the limitation of a deterministic policy class. Besides diffusion policies, training policies to achieve diverse outputs has long been of interest, especially in the maximum entropy framework (Celik et al., 2025; Haarnoja et al., 2018). Here, we introduce entropy regularization for diffusion policies to increase the diversity of sampled sequences. Note that, while building upon ideas from maximum entropy RL research, our method is not a maximum entropy RL method, as we do not combine the entropy term with the Q-value to be maximized. Instead, we regularize the entropy of the diffusion policy for a single action selection.

## 4 Sequential Diffusion-Policies for Antibody Sequence Design

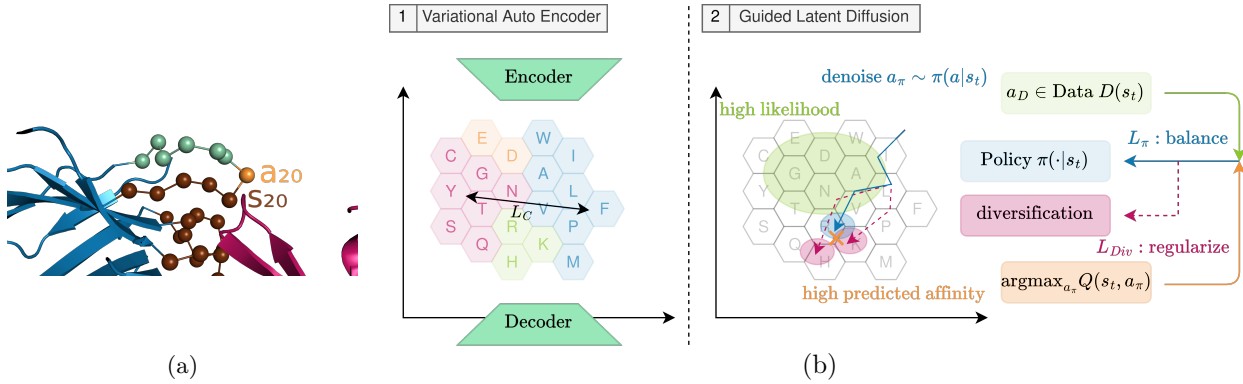

(a)                                                                 (b)

Figure 1: (a) Antibody Sequence Design Task: For each of the $L$ residues in the antibody CDRs, shown as beads, an AA has to be assigned to minimize the free energy between the antibody (blue and beads) and the antigen (red). We formulate the task as an iterative MDP where the AA for the current residue (shown in yellow) has to be assigned conditioned on all residues designed so far (shown in brown). Starting from an empty sequence $s_0$, an RL policy $\pi_\theta$ iteratively designs each AA $a_t$ given the previous AAs $s_t$ until the entire sequence $s_{L-1}$ is designed. (b) Antibody Sequence Design Agent: (b.1) We use a VAE to encode AAs into a two-dimensional latent space. Optionally, a contrastive loss $L_C$ (Section 4.1) can be used to group AAs based on biophysical priors, such as their side chain properties. (b.2) During inference, we sample a two-dimensional latent AA representation $\boldsymbol{a}_t \sim \pi_\theta(\boldsymbol{a}_t|s_t)$ from the diffusion policy given an incomplete AA sequence and decode it using the frozen VAE. During training, $L_\pi$ (Section 4.2) balances the policy $\pi_\theta$ between generating AAs with high likelihood given the training dataset $D$ and AAs that maximize a learned Q-function $Q^\pi$, which predicts sequence affinity to a given antigen. An optional entropy maximization $L_{Div}$ (Section 4.3) can be used to increase the diversity of generated AAs.

The objective of our method, SeqDiff+QL, summarized in Figure 1, is to generate high-affinity CDR sequences while remaining in the proximity of the empirical antibody sequence distribution defined by a dataset $D$ of sequence-affinity pairs. We approach the design of a sequence $s$ in a stepwise manner. Con-

ditioned on the incomplete sequence $s_t$, the policy $\pi_\phi(\boldsymbol{a}_t|s_t)$ is used to iteratively generate AAs which are concatenated with $s_t$ to extend the sequence. As the generated sequences should be novel and diverse, we represent the policy $\pi_\phi$ using a continuous latent diffusion model, which can model complex distributions and generate diverse, high-quality samples (Ho et al., 2020; Murphy, 2023; Wang et al., 2023).

### 4.1 Encoding Biophysical Properties through Continuous Amino Acid Representations

To apply continuous diffusion policies to the task of discrete AA generation, we encode each AA as a two-dimensional latent embedding $\boldsymbol{a}$ using a VAE (Kingma & Welling, 2014). To train the VAE, each AA $a$ is represented as a one-hot vector and mapped to a two-dimensional latent $\boldsymbol{z} = e_\omega(a) \sim \mathcal{N}(\mu_\omega^a, \sigma_\omega^a)$ using the encoder network $e_\omega$. The decoder network $d_\psi(\boldsymbol{z})$ then maps the latent vector $\boldsymbol{z}$ back to a probability distribution over discrete AAs. The VAE is trained end-to-end by minimizing the Binary Cross Entropy (BCE) loss between the input $a$ and the decoder's output. Additionally, the distribution of latent variables $\boldsymbol{z}$ is regularized to minimize the Kullback–Leibler (KL) divergence $D_{KL}$ to the Gaussian distribution $D_{KL}(\mathcal{N}(\mu_\omega^a, \sigma_\omega^a)||\mathcal{N}(0, \mathbf{I}))$. This promotes a dense and continuous latent space while preventing discrepancies between the VAE latent space and the Gaussian noise $x^N \sim \mathcal{N}(x^N; 0, \mathbf{I})$ used in the diffusion process.

In an arbitrarily organized latent space, small deviations in the non-deterministic diffusion process can cause large functional changes in the generated AA. To alleviate this, we induce biophysical priors in the generative process such that AAs are grouped by functional similarities induced by their side-chain polarity, as visualized in Figure 1(b). See Section A.6 in the appendix for details on our chosen grouping. To incorporate the grouping of AAs by biophysical properties, we added a supervised contrastive loss to the VAE training objective (Khosla et al., 2020). Specifically, the contrastive loss is given by $L_C(a) = -log[\frac{1}{|G(a)|} \sum_{p \in G(a)} \frac{\exp(\boldsymbol{z}_a \cdot \boldsymbol{z}_p)/\tau}{\sum_{a' \in A \setminus a} \exp(\boldsymbol{z}_a \cdot \boldsymbol{z}_{a'})/\tau}]$, where $A$ is the set of all AAs, $G(a)$ represents the subset of AAs belonging to the same functional group as $a$, cosine similarity over latent representations $\boldsymbol{z}$ is represented by $\cdot$, and $\tau$ is a temperature hyperparameter. This loss maximizes the similarity between AAs in the same group and minimizes it between groups. The loss function of the VAE is then given as $L(a) = \text{BCE}(a, d_\psi(e_\omega(a))) + \text{KL}(\mathcal{N}(\mu_\omega^a, \sigma_\omega^a)||\mathcal{N}(0, \mathbf{I})) + L_C(a)$. Note that the VAE is pre-trained and kept frozen during the training of the diffusion process.

Compressing 20 amino acids into a two-dimensional latent is a relatively tight bottleneck. See Section A.11 in the appendix for a discussion and ablation on this hyperparameter choice.

### 4.2 Guiding Diffusion Policies using Reinforcement Learning

Our training dataset includes up to 20 possible AAs for extending an incomplete sequence and is highly multimodal. This further motivates the use of continuous diffusion models as generative RL policies, which proved more effective than other training paradigms when dealing with multimodal data (Park et al., 2025; Wang et al., 2023). Thereby, the diffusion policy $\pi_\theta$ is trained to achieve a balance between two objectives: generating latent vectors representing AAs with high likelihood given a dataset $D$ and generating AAs maximizing a learned Q-function.

The loss function corresponding to the first objective, referred to as the BC loss, is a slight adaptation of the standard loss function for continuous diffusion models given in Section 2.2. In particular, as we generate sequences stepwise, one AA $a$ after the other, we condition the diffusion model on the sequence $s$ of AAs generated so far. The resulting BC loss is given by $L_{BC}(\theta) = \mathbb{E}_{n \sim \text{Unif}(1,N), \epsilon \sim \mathcal{N}(0,\mathbf{I}), (s,a) \sim D}[||\epsilon - \epsilon_\theta(\sqrt{\bar{\alpha}^n}\boldsymbol{a} + \sqrt{1 - \bar{\alpha}^n}\epsilon, s, n)||^2]$. Simply put, this loss function trains the model to reconstruct the latent representation $\boldsymbol{a}$ for the next $a$ in a sequence from a noisy sample conditioned on an incomplete sequence $s$ from the dataset $D$.

Using policy improvement, the diffusion policy can be optimized toward high-affinity sequences while remaining close to the data distribution. We follow prior work (Park et al., 2025; Wang et al., 2023) and maximize the Q-function $Q(s, \boldsymbol{a}^0)$ given an incomplete sequence $s$ and a latent action $\boldsymbol{a}^0$ generated by the policy $\pi_\theta$. As $\boldsymbol{a}^0$ is generated using the reverse process of the diffusion model $\pi_\theta$, the gradient of $Q_\phi(s, \boldsymbol{a}^0)$ with respect to $\boldsymbol{a}^0$ is propagated through the diffusion model's reverse process, thereby guiding

the selection of actions with a high Q-value given the current state $s$. In contrast to guided discrete diffusion (Gruver et al., 2024), continuous diffusion policies directly support this gradient propagation without any architectural modifications. By combining the $L_{BC}$ with Q-maximization, we define the policy loss $L_\pi$ as $L_\pi(\theta) = (1 - \eta) \cdot L_{BC}(\theta) - \eta \cdot \mathbb{E}_{s \sim D, \boldsymbol{a}^0 \sim \pi_\theta(\boldsymbol{a}^0|s)}[Q_\phi(s, \boldsymbol{a}^0)]$. The combination of the likelihood term $L_{BC}$ and Q-maximization can be interpreted as regularized policy improvement in an offline setting: the diffusion likelihood term keeps the policy close to the empirical sequence distribution, effectively acting as a soft trust region around the behavior data, while the value term drives affinity improvement. A similar combination has proven effective in many offline RL domains (Fujimoto & Gu, 2021). Furthermore, using a Q-function for guidance in sequence design is particularly promising, given its ability to stitch together improved sequences from suboptimal ones and its excellent performance in states requiring specific actions (Kumar et al., 2022).

The Q-function $Q_\phi$, implemented as clipped double Q-learning (Fujimoto et al., 2018), is trained to minimize the so-called TD-error: $L_{Q_i} = \mathbb{E}_{(s_t, a_t, s_{t+1}) \sim D, \boldsymbol{a}^0_{t+1} \sim \pi'_\theta(\boldsymbol{a}^0_{t+1}|s_{t+1})}[||R(s_t, a_t) + \min_{j=1,2} Q_{\phi'_j}(s_{t+1}, \boldsymbol{a}^0_{t+1}) - Q_{\phi_i}(s_t, \boldsymbol{a}_t)||^2]$, where subscripts $t$ indicate the trajectory index (AA position). To prevent overestimation, we implement the Q-function as a categorical distribution as proposed by Farebrother et al. (2024). During training, the diffusion policy $\pi_\theta$ and the Q-function $Q_\phi$ are updated alternately. We refer to our algorithm, without any improvements presented in the following sections, as SeqDiff+QL. However, this algorithm is lacking a mechanism to control the diversity of generated sequence distributions, which we describe in the following section.

### 4.3 Increasing Diversity of generated Sequences

The ability to generate a diverse set of candidate sequences is of high importance for biological screening, due to the diverse nature of antibody-antigen interaction as well as the limited ability of in silico simulations to mimic real-world interaction (Jain et al., 2022). Finetuning Diffusion Models with RL can, however, lead to reduced diversity and mode collapse (Barceló et al., 2024).

To counteract this, we add a entropy regularization to increase the diversity of samples generated. Specifically, we introduce the auxiliary loss $L_{Div}(\theta) = \mathbb{E}_{p_\theta}[\log \frac{p_\theta(\boldsymbol{a}^{0:N}|s)}{q(\boldsymbol{a}^{1:N}|\boldsymbol{a}^0)}]$, where $\boldsymbol{a}^0$ is sampled from $p_\theta(\boldsymbol{a}^0|s)$. This auxiliary loss maximizes a lower bound on the marginal entropy of the reverse process $p_\theta$ Celik et al. (2025), by minimizing the KL divergence $D_{KL}(p_\theta(\boldsymbol{a}^{0:N}|s)||q(\boldsymbol{a}^{1:N}))$ between forward and learned backward diffusion process representing the policy $\pi_\theta(\boldsymbol{a}|s)$. Integrating this into the policy loss as $L_{\pi+Div}(\theta) = L_\pi(\theta) + \rho \cdot L_{Div}(\theta)$ allows increasing diversity in addition to preserving the trust-region constrained induced by the dataset likelihood.

We adopt this particular regularizer because, in our setting, the failure mode of Q-guided diffusion training is mode collapse onto a small number of high-Q-value sequences: the policy stops covering the conditional support and starts repeating a few amino acids per position. Maximizing a lower bound on the per-step marginal entropy of $\pi_\theta(\boldsymbol{a}|s)$ directly penalizes that collapse, while remaining agnostic to the discrete sequence space and to the specific oracle. We do not claim it is the uniquely correct regularizer for sequence diversity. Rather, it is one principled, application-agnostic choice; downstream applications could substitute a more domain-specific term, for example a KL regularizer towards the empirical CDR amino-acid distribution from natural antibody repertoires, when such priors are available and trustworthy. We view the entropy term as a baseline diversity regularizer compatible with the diffusion-policy formulation, not as a theoretical optimum.

### 4.4 Identifying promising generated Sequences

To reduce cost, it is advantageous to select the most promising generated candidates before real-world evaluation. Vázquez Torres et al. (2025) use Alphafold2 and Rosetta metrics for this task. Recall from Section 2.1 that for a sequence of length $L$, the Q-value $Q_\phi(s_{L-1}, \boldsymbol{a}_{L-1})$ of $s_{L-1}$ and the last amino acid $\boldsymbol{a}_{L-1}$ is trained to predict the sequence's affinity. Thus, we use the learned Q-values as a principled scoring function, sorting generated sequences by their predicted free energy and identifying high-affinity sequences without additional training or auxiliary methods. This can be used as a post-generation selection scheme.

# 5 Experiments

In this section, we present experimental results across two distinct benchmark environments and multiple relevant data distributions. We design our experiments to reflect realistic variations in desired sequence length, variations in training data distribution, and variations in evaluation procedures. All experiments are carried out over five seeds. Note that the datasets and source code will be made publicly available.

## 5.1 Antibody Sequence Design Tasks

There are multiple tools available to estimate binding affinity, or the inversely proportional free energy, which could be used to create antibody sequence design tasks. However, many of them are available only as web servers, come with high resource demands, and provide no, outdated, or only partial code release (Abbasi et al., 2020; Jain et al., 2022; Li et al., 2024; Myung et al., 2021; Romero-Molina et al., 2022; Xue et al., 2016; Yang et al., 2023). For our analysis, we chose two tools, Absolut! (Robert et al., 2022) and Rosetta (Simons et al., 1997), which can be installed locally and are relatively lightweight, taking 6.2 and 10.2 seconds per sample evaluation on consumer hardware, respectively. We create two antibody sequence design tasks, where the goal is to maximize the affinity / minimize free energy estimated using Absolut! or Rosetta, respectively. Here, we give a short description of the task details. See Section A.1 in the appendix for an in-depth description of both tasks. In our main experiments, we focus on a single antigen per oracle: SARS-CoV Spike Receptor-Binding Domain (PDB ID 2DD8) in Absolut! and the B-cell maturation antigen (PDB ID 8HXQ) in Rosetta. To probe cross-antigen generalization, we additionally evaluate every method on a second antigen per oracle: Human CD38 (PDB ID 3RAJ) in Absolut! and the PaaR2 N-terminal domain (PDB ID 8C3K) in Rosetta. A summary is given in Section 5.4.3, with full tables in Section A.8.

**Absolut!** The goal is to generate sequences of length $N = 11$ AAs, representing the CDRH3 region of an antibody, to maximize binding affinity based on a 3D lattice-discretized representation of the CDRH3 and a chosen antigen and the Miyazawa-Jernigan energy potential (Miyazawa & Jernigan, 1999). In our study we choose the antigens SARS-CoV Spike Receptor-Binding Domain (PDB ID 2DD8) and human CD38 (PDB ID 3RAJ), which have shown to be challenging for sequence design in prior work (Vogt et al., 2023; Cowen-Rivers et al., 2022). We curated three diverse sequence distribution datasets, referred to as *random*, *natural*, and *expert*, reflecting distributions as they could occur in real-world applications. Dataset sizes range from 2167 to 2753 unique sequences.

**Rosetta** We use the Rosetta Energy Function *REF15* (Alford et al., 2017) to estimate the free energy between a nanobody (single-domain antibody) and the B-cell maturation antigen (PDB ID 8HXQ) based on their 3D structure. The goal is to redesign all CDRs of the nanobody (CDR1, CDR2, and CDR3), to minimize free energy. This leads to a total of $N = 28$ AAs to be designed. We curated a random dataset, comprising 2448 sequences, and an expert dataset. To create the expert sequences, we recreated the computational sequence design pipeline as utilized by Dauparas et al. (2022) and Vázquez Torres et al. (2025), which was used to generate sequences successfully verified in multiple real-world experiments (Dauparas et al., 2022; Vázquez Torres et al., 2025). In this approach, we first sampled 1000 CDR backbone structures using RFdiffusion (Watson et al., 2023) conditioned on the nanobody and antigen structure. We then applied Protein-MPNN (Dauparas et al., 2022) to sample a total of 2483 unique sequences likely to fold into the respective structures. As a second antigen we selected the PaaR2 N-terminal domain (PDB ID 8C3K), which contains a total of 20 CDR residues. A small fraction of Rosetta evaluations yield unstable extreme energies, which we discard via an energy threshold (see Section A.1); we assess the sensitivity of our conclusions to this choice in Section A.9.

## 5.2 Evaluation Process

The main goal in the antibody design task, and thus also the reward agents are trained to maximize, is the binding affinity of an antibody towards an antigen as described above. Similar to Jain et al. (2022), we evaluate this via the mean free energy of the top 100 candidates generated by each method. We choose to

evaluate the top 100 instead of the entire set of generated sequences, as finding a few or even a single good binder often suffices, and it is not of primary interest to optimize the mean affinity of generated sequences.

Besides the primary task of maximizing the binding affinity towards a target, it is of large interest to generate novel and diverse candidate sequences. To evaluate these properties, we utilize the definition of diversity and novelty proposed by Jain et al. (2022): $Diversity(D_{gen}) := \frac{\sum_{x_i \in D_{gen}} \sum_{x_j \in D_{gen} \setminus \{x_i\}} d(x_i, x_j)}{|D_{gen}|(|D_{gen}|-1)}$ and $Novelty(D_{gen}) := \frac{\sum_{x_i \in D_{gen}} \min_{x_j \in D} d(x_i, x_j)}{|D_{gen}|}$, where $D$ represents the training dataset and $D_{gen}$ the dataset of generated sequences, while $d(\cdot, \cdot)$ is the Levenshtein distance quantifying the difference between two sequences (Miller et al., 2013). These auxiliary measures provide insight into the average number of pointwise mutations in the sequence relative to other sequences in the generated dataset $D_{gen}$ (diversity) and their closest relative in the original dataset $D$ (novelty).

In our experiments, we run the generative process of all evaluated methods until we receive 500 unique novel sequences or reach a maximal budget of 6144 generated sequences. We then compute novelty and diversity over the set of unique and novel sequences generated, and additionally count the number of duplicates in the generative process (either with sequences in the dataset or with previously generated sequences by the same method). This way, we can observe when an algorithm can be used to generate a diverse but very limited number of unique sequences.

## 5.3 Baselines and Implementation

We introduce a set of baselines containing classical RL algorithms, constructing sequences step by step, and Diffusion Models, generating sequences simultaneously. To facilitate a fair comparison, we use the same network architecture for shared components in all algorithms. Here, we present only a brief overview. For more details, see Section A.5 in the appendix. The baselines consist of:

**Behavior Cloning (BC)**: Sequential BC policy, trained using cross-entropy loss to estimate the next action (AA distribution) conditioned on the current state. Suited for diverse generation and offline learning, but not for objective-driven design.

**Behavior cloning + Q-learning (BC+QL)**: Sequential policy, combining BC with Q-learning (QL) to balance staying close to the training dataset and maximizing the affinity. This combination was previously successfully applied in continuous action settings and showed remarkable performance in the offline RL setting (Goecks et al., 2019; Fujimoto & Gu, 2021; Nair et al., 2021). We extend it to the discrete action setting. Suited for diverse, objective-driven design and offline learning.

**Simultaneous Diffusion (SimDiff)**: Simultaneous BC diffusion policy, implemented using Denoising Diffusion Probabilistic Model (DDPM) (Ho et al., 2020), generating all AAs in the sequence at once. Suited for diverse generation and offline learning, but not for objective-driven design.

**Sequential Diffusion (SeqDiff)**: Sequential BC diffusion policy, which models the training data distribution by sequentially sampling AAs. Equivalent to only using the $L_{BC}$ component of our method SeqDiff+QL. Suited for diverse generation and offline learning, but not for objective-driven design.

**GFlowNet**: Sequential policy, combining generative flows with model-based data generation. Proposed by Jain et al. (2022) to design biological sequences.

We carried out a hyperparameter search to identify suitable parameters for our method and all baselines. For details, see Section A.4 in the appendix.

## 5.4 Results

We first present an analysis of our method without modifications, such as entropy regularization and biophysical priors, and baselines in both the Absolut! and Rosetta environments, followed by an ablation study for all modifications. We choose a visual presentation in this section, for tabular results and significance tests see Section A.2 in the appendix.

### 5.4.1 Absolut!

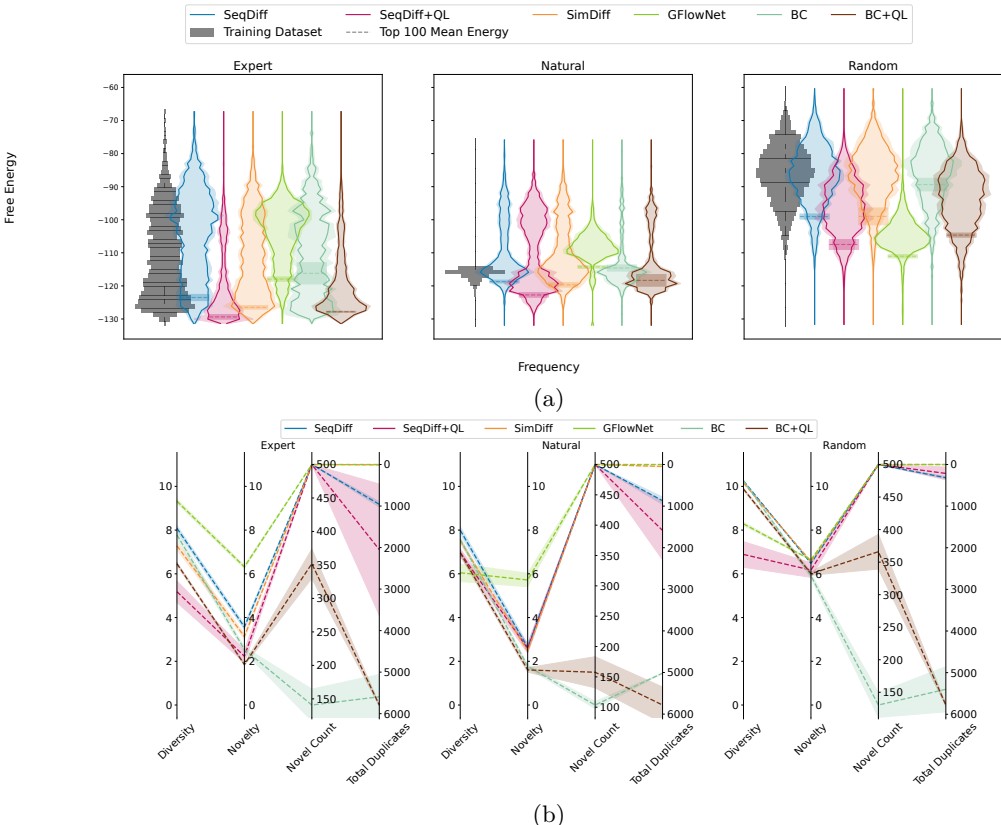

(a)

(b)

Figure 2: (a) Free energy distributions of the top 100 unique, novel generated sequences in the Absolut! task. Dataset distributions are visualized as histograms. Lower free energy is preferred. (b) Auxiliary Metrics: Diversity, Novelty, Novel Count and Total Duplicates, defined in Section 5.2, in the Absolut! task.

In Figure 2, we visualize the free energy distributions and auxiliary metrics on the Absolut! task.

For the methods without objective-driven design, we observe that BC and SimDiff do model the energy distribution of the expert and random dataset well, while SimDiff exhibits reduced performance on the natural dataset. Further, SeqDiff exhibits reduced modeling capability compared to these methods on the expert and natural data distribution, generating an increased amount of low-affinity sequences. This can be attributed to a higher randomness in the generative process, which is also supported by the increased diversity and novelty. This shows that while modeling of the distribution via sequential design, as indicated by BC, and via diffusion models, as indicated by SimDiff, the combination of both leads to decreased modeling capability.

For methods employing an objective-driven design, we observe strong improvements in mean affinity of the top 100 sequences in BC+QL and SeqDiff+QL trained on the expert and random datasets, compared to their non-QL counterparts. Thereby, the mean affinity in the top 100 sequences is significantly higher for SeqDiff+QL than for all other methods, except for the random dataset. In contrast, we observe that GFlowNet does not generate many high-affinity sequences on the expert and natural dataset but excels on the random dataset, where it achieves significantly higher affinity scores than all other methods. Note that to achieve good performance on the natural dataset, the $\eta$ hyperparameter of our method SeqDiff+QL had to be decreased, highlighting the importance of dataset constraints in this setting. When trying to increase the focus on high-affinity sequences in GFlowNet via the corresponding hyperparameter, we observed an overfitting to model-bias leading to decreased diversity but not improved energy.

When analyzing the auxiliary metrics diversity, novelty, and the number of novel generated sequences and duplicated sequences, we observe that BC and BC+QL while generating a set of novel, unique sequences with comparable novelty and diversity generate significantly more duplicated sequences (either with the training dataset or previously generated sequences) and exceed the generative budget of 6144 sequences before reaching the desired amount of 500 novel, unique sequences. This indicates that while these non-diffusion methods can be adapted to perform objective-driven design, their usability to generate extended sets of novel and diverse sequences is limited. GFlowNet, on the other hand, did not create any duplicated sequences and exhibited high diversity and novelty. We further observe in BC+QL and SeqDiff+QL that the use of QL leads to a decrease in diversity and novelty. We attribute this to a decline in multi-modality, which has been previously observed for RL-based finetuning of diffusion models (Barceló et al., 2024). In Section A.10, we present a deeper analysis of this mode collapse and further show that $\eta$ acts as a direct lever on the resulting distribution shift. Sweeping $\eta$, we report a set of distribution-shift indicators (Diversity, Novelty, Mean Energy, Earth Mover's Distance (EMD) of the generated energy distribution, Latent Distance, and BC loss) together with the position-wise amino-acid logo plots as a direct sequence-space comparison. As $\eta$ grows, the policy concentrates on a number of high-Q sequences and shifts towards a high-affinity region that remains close to the data distribution, with the cost paid in novelty rather than as a departure from the training manifold.

Note that while we follow Jain et al. (2022) in the analysis of novelty and diversity, we observe limitations of the metric in the Absolut! setting. High-affinity sequences in Absolut! share similar patterns/modes and are thus less diverse per se. We carried out a deeper analysis of the relationship between affinity and diversity in section A.7 and show that diversity decreases monotonically as binding strengthens. Thus, the reduced diversity of high-performing agents partially reflects an intrinsic property of the high-affinity region rather than the agent alone and diversity measures should always be regarded in the context of the corresponding energies. Normalizing diversity of sequences with respect to the energy distribution they are sampled from would be desirable, but is infeasible in our setting as the large sequence-space does not allow for exhaustive sampling. Normalizing by diversity references derived from policy-generated sequences would be highly biased, while the large murine reference set, comprising 6.9 million CDRs, provided by Robert et al. (2022) is too sparse outside the densely-sampled central range to anchor the affinity regime our objective-driven methods reach. We therefore retain the metric of Jain et al. (2022) for cross-paper comparability and report diversity paired with the corresponding energies.

### 5.4.2   Rosetta

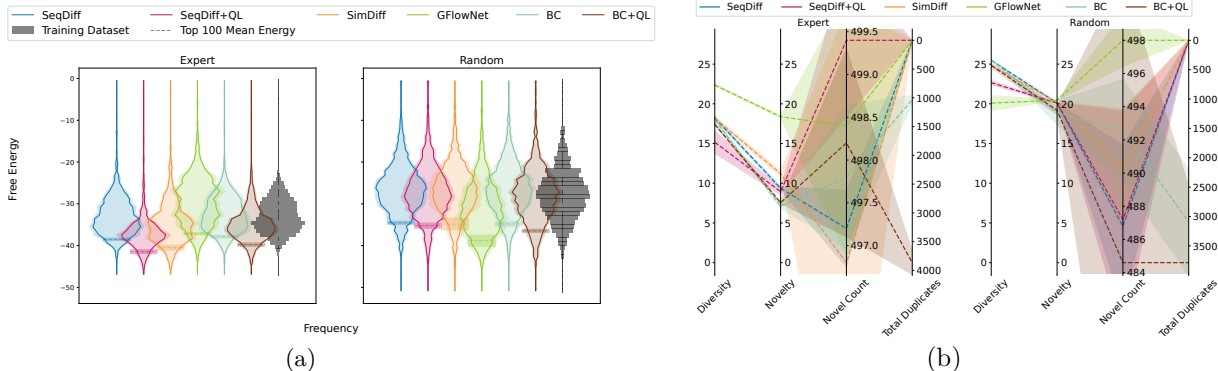

Figure 3: (a) Free energy distributions of the top 100 unique, novel generated sequences in the Rosetta task. Dataset distributions are visualized as histograms. Lower free energy is preferred. (b) Auxiliary Metrics: Diversity, Novelty, Novel Count and Total Duplicates, defined in Section 5.2, in the Rosetta task.

In Figure 3, we visualize the free energy distributions and auxiliary metrics on the Rosetta task. Similar to results in Section 5.4.1, we observe an improved affinity through QL in comparison to non-QL counterparts and a significantly higher performance of SeqDiff+QL than all other methods on the expert dataset. Further, we again observe a low performance of GFlowNet on the expert dataset but high performance on the random

dataset, where it achieves significantly better results. In contrast to the Absolut! task, we also observe a comparably small improvement through QL for SeqDiff+QL on the random dataset.

With respect to auxiliary metrics, we again generally observe a decrease in diversity and novelty for QL-based agents. Also GFlowNet exhibited a reduced novelty on the random dataset. On the Absolut! dataset, we did not observe any issues with duplicates being generated, which can be attributed to a higher length of generated sequences ($L = 28$ vs $L = 11$ in Absolut!). Note that due to outlier removal from Rosetta evaluation, see Section A.1 for details, the number of granted sequences can be slightly below 500.

Lastly, we observe that most methods top 100 as well as the overall mean affinity improve upon the expert dataset. This shows potential to use such objective-driven methods to improve upon candidate distributions generated using RFdiffusion and Protein-MPNN, which currently represent a gold standard for protein engineering (Dauparas et al., 2022; Vázquez Torres et al., 2025).

### 5.4.3 Second-antigen evaluation

To probe cross-antigen generalization, we evaluate every method on a second antigen per oracle: 3RAJ for Absolut! and 8C3K for Rosetta. The full per-method tables, with the same five seeds and identical hyperparameters as the first antigen, are given in Section A.8 (Table 3 for Absolut! and Table 4 for Rosetta). On the second Absolut! antigen, the qualitative ordering carries over: SeqDiff+QL attains the best top-100 free energy on the expert dataset and improves substantially over SeqDiff on random, while GFlowNet dominates on random and on this antigen also on natural. On the second Rosetta antigen, the picture is more nuanced. The expert dataset is itself already very high-affinity (mean −40.5 Rosetta Energy Units (REUs)). Several baselines, including non-guided SeqDiff and SimDiff, are statistically tied with SeqDiff+QL on top-100 affinity, and SeqDiff+QL retains a small lead but no longer separates from SeqDiff as cleanly as on 8HXQ. On random data, BC+QL performs best on this antigen, while SeqDiff+QL still improves over the unguided baselines. The qualitative claims, Q-guidance helps and reduces diversity, are antigen-robust; the strict ranking of the best objective-driven method is antigen-dependent. We refer the reader to Figure 8 and Figure 9 in the appendix for the corresponding free-energy distributions and auxiliary metrics.

### 5.4.4 On the GFlowNet inconsistency

In our experiments we notice an inconsistency in the performance of GFlowNet across datasets and oracles. Generally, GFlowNet performs well on the random dataset but underperforms on the expert and natural datasets, which is consistent across both oracles, with the exception of the second Absolut! antigen, where GFlowNet performs well on both random and natural. We attribute this to the role of the proxy-oracle employed by Jain et al. (2022). During training, the GFlowNet policy is trained on a mixture of samples from the training data and the proxy-oracle. On the one hand, if the proxy-oracle aligns well with the true oracle on samples outside the training distribution, this can be beneficial and guide the policy to novel high-affinity sequences. On the other hand, if the proxy-oracle is misaligned with the true oracle, this can lead to model bias and guide the policy towards a region of sequence space that is erroneously predicted to have high affinity, leading to low true affinity. A particular case that highlights this bias well is the performance of GFlowNet in the second analysed antigen (PDB ID 3RAJ) in Absolut! natural and random datasets. In this case, the proxy-oracle captured a bias in the Absolut! oracle, which led to an extraordinary performance of GFlowNet in Figure 8. However, the policy did also overfit to a few hydrophobic AAs as visible in the corresponding logo plots in Figure 10, which is a direct consequence of the proxy bias.

### 5.4.5 Entropy Regularization, Biophysical Priors and Q-value based Filtering

In the following, we present ablation studies to show the effect of the introduced entropy regularization, the addition of biophysical priors, and identifying high-affinity sequences through learned Q-functions.

**Entropy Regularization** In our previous experiments on Absolut! and Rosetta tasks, we observed that QL led to increased affinity but reduced diversity. A possible cause for this effect is a reduced diversity in the generative process. In this section, we will analyze the effect of entropy regularization as a way to counteract this phenomenon. In Figure 4 we present the Pareto front with respect to affinity and diversity of

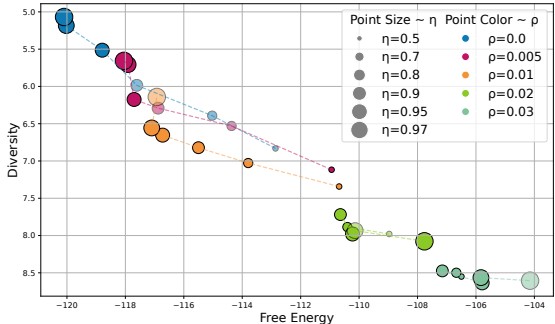

Figure 4: Pareto front over $\eta$ and $\rho$ on the Absolut! expert dataset. Agents using entropy regularization are Pareto-dominant over non-regularized configurations on large areas of the energy-spectrum, giving both higher affinity and diversity. Very high affinity scores can only be reached with non-regularized settings with high weight on Q-guidance.

novel generated sequences on the Absolut! expert dataset. We observe that there exists a range of very high affinity, which is only reachable with high $\eta$ and no entropy regularization ($\rho = 0.0$, shown in blue). However, as in the previous experiments, these come with the downside of having reduced diversity. By utilizing our introduced entropy regularization, we can reach a set of configurations (shown in red and orange) which are Pareto-dominant over configurations without regularization in the high-affinity range, giving both a higher affinity and diversity compared to non-regularized agents. As a point of reference, a free energy below -114.83 is better than 99.9% of the 6.9 million murine CDRH3 sequences tested in the Absolut! publication (Robert et al., 2022). This shows that entropy regularization effectively allows for counteracting the decrease in diversity through guidance, even for high-affinity ranges.

**Biophysical Priors and Q-value based filtering** As the integration of biophysical priors is independent of the Q-learning mechanism, and Q-value-based filtering affects the entire sequence distribution, we carry out this analysis on a fixed $\eta = 0.9$ setting and analyze the mean energy of all generated sequences instead of the top 100 sequences. Note that the same effect is visible using other configurations and the top 100 sequences. In Figure 5(a), we visualize the effect of integrating biophysical priors into the latent space via a contrastive loss. We observe that AAs become clustered according to their group, representing their side-chain properties. Especially in the Absolut! environment, this helps to group high-valued hydrophobic AAs.

As a result, visualized in Figure 5(b), the mean affinity of generated novel sequences increases on all Absolut! environments. In the Rosetta environment, the positive effect is not observable. This can be attributed to our chosen prior coinciding with the Miyazawa-Jernigan energy potential (Miyazawa & Jernigan, 1999) used in the Absolut! reward function, but less so with the Rosetta energy function. See Section A.7 for a detailed analysis.

Filtering generated sequences according to their estimated Q-values mostly allows identifying good sequences, as indicated by the higher affinity of the top 50% sequences sorted by estimated Q-values, shown in Figure 5(b). These observations, like in the previous experiments, do not hold for the random Rosetta dataset. For this particular dataset, we observe a very low correlation between learned Q-values and real binding values (p=0.015). We hypothesize that this could be due to low signal in the random data and low coverage of only 2448 sequences, while designing 28 residues and thus covering a design space of $20^{28}$ sequences. Lastly, we observe that for all evaluations except for the one on random Rosetta data, the incorporation of priors and the filtering are compatible, leading to even higher affinity scores if used in combination. To quantify this across settings, we report the Spearman rank correlation between learned Q-values of SeqDiff+QL, BC+QL, and GFlowNet and the actual oracle energy of novel generated sequences, on both antigens and all dataset distributions, in Table 7 of Section A.12. There, we observe that for the random dataset in the Rosetta environment, the correlation is indeed lower than in all other settings, which is consistent with the observed lack of improvement through Q-value-based filtering.

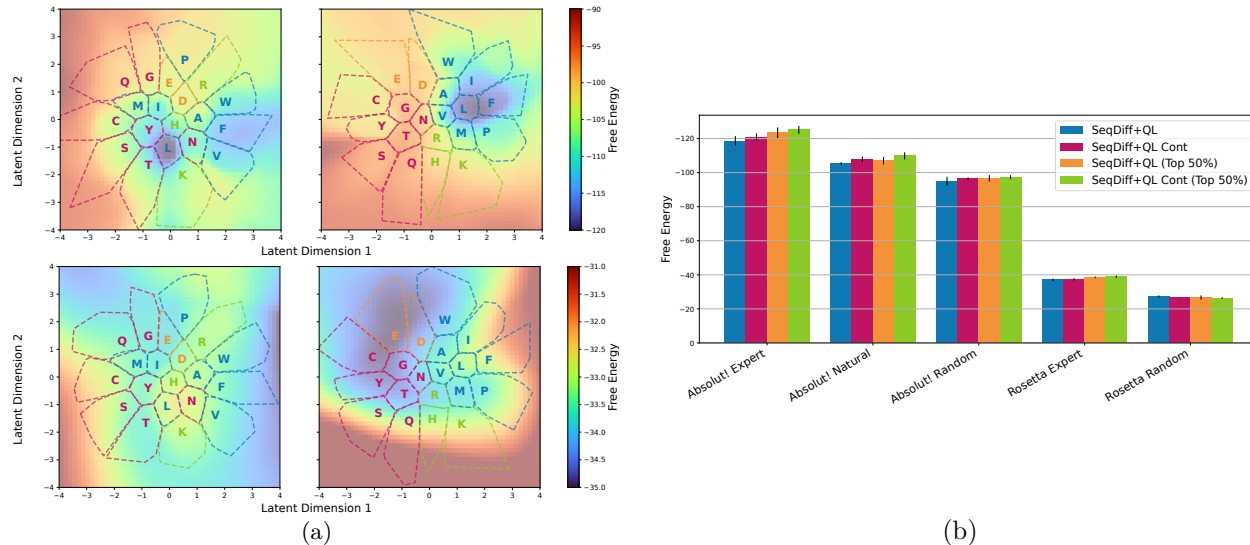

Figure 5: (a) Learned latent spaces in the Absolut! expert (top) and Rosetta expert (bottom) task. Convex hulls indicate areas encoding for a given amino acid. The left plot shows a random encoding, while the right plot highlights the effect of contrastive losses conditioned on amino acid properties. Heatmaps indicate learned Q-values. (b) The effect of integrating biophysical priors into the VAE latent space (Cont) and filtering the top 50% of sequences according to learned Q-values (Top 50%) in our SeqDiff+QL algorithm.

## 6 Conclusion

We presented SeqDiff+QL a novel diffusion-based RL method for antibody sequence design, and evaluated it on multiple antibody sequence design tasks with varying training data distributions, sequence lengths, and evaluation functions using the Absolut! and Rosetta software. We showed that our method is applicable for objective-driven design of novel sequences with improved oracle-predicted affinity, and significantly outperforms a diverse set of baseline methods, comprising classical RL methods, diffusion methods, and GFlowNet, on a majority of the in silico tasks we consider. The comparison to GFlowNet highlights room for improvement of our method on some data distributions, which potentially could be alleviated using a model-based approach similar to GFlowNet. For all objective-driven methods, we observed an increase in oracle-predicted affinity, but also a measurable decrease in sequence diversity and novelty. Our proposed entropy regularization partially mitigates this tradeoff, recovering some diversity in part of the affinity range while the very highest predicted-affinity regime still requires non-regularized configurations and the corresponding diversity cost. We further demonstrated how learned Q-values can be used to identify promising candidates in the set of generated sequences and that biophysical priors in the diffusion process can improve the affinity of generated sequences if the priors coincide with those present in the evaluation method. All evidence in this paper is computational. Translating in silico affinity gains into real binding or therapeutic value would require wet-lab validation, which we leave to future work; in particular, the method should be viewed as a candidate generator that ranks well under specific oracles rather than as a demonstrated path to real-world antibody discovery.

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

# A    Appendix

## A.1    Evaluation Tasks

### A.1.1    Absolut!

The Absolut! software can be used to estimate the free energy of a sequence of length $L = 11$ AAs, representing the CDRH3 region of an antibody, based on a 3D lattice-discretized representation of the CDRH3 and the antigen and the Miyazawa-Jernigan energy potential (Miyazawa & Jernigan, 1999). The lattice-based representation allows for faster approximation, leading to a computation time of 6.2 seconds for a single sample. For the Absolut! design task, we choose the SARS-CoV Spike Receptor-Binding Domain (PDB 2DD8_S) as the antigen, which in previous publications was one of the most challenging targets evaluated (Cowen-Rivers et al., 2022; Vogt et al., 2023).

We curated three training sequence distributions with corresponding energy values reflecting distributions as they could occur in real-world applications. The first distribution comprises a set of 2500 randomly generated sequences. The second set contains 2753 murine CDRH3 sequences, which were categorized as good but not exceptional binders in the Absolut! publication Robert et al. (2022) (specifically the top 0.01% to 0.1% of 6.9 million tested murine CDRH3 sequences. The final distribution, comprising 2167 sequences, was gathered during the exploration phase of an online Q-learning agent, similar to those described by Vogt et al. (2023). We refer to the three datasets as *random*, *natural*, and *expert*. These datasets reflect scenarios that could occur in an application of our method on distributions from random essays, enrichment data, and active learning AI pipelines.

### A.1.2    Rosetta

The Rosetta software (Simons et al., 1997) is an established software for many biomolecular modeling tasks. To create the Rosetta design task, we utilize the Rosetta Energy Function *REF15* (Alford et al., 2017) to estimate the free energy in a nanobody-BCMA complex structure (PDB: 8HXQ). The comparably small antibody and antigen in this complex allow a relatively fast evaluation (10.2 seconds per sample) despite the overall high cost of the Rosetta software. The goal of this task is to design all three CDRs sequences

in the nanobody such that the free energy is minimized. To evaluate the energy of a designed sequence, we execute the following steps. First, we relax the original complex using Rosetta's FastRelax protocol to reach a low-energy state. Using Rosetta's MutateResidue protocol, we then mutate the AAs in the complex to the newly designed sequences. To remove any clashes introduced throughout the mutation, we apply FastRelax again to any mutated residue and any residues within a 10 Å neighborhood. Lastly, we utilize Rosetta's InterfaceAnalyzerMover to compute the binding affinity (measured as dG_separated). Typically, this roughly leads to estimated free energy values of -1 to -50 REUs for 2500 random sequences. However, in ∼2% of evaluated sequences, the estimated binding energy is far outside those we typically observe for this complex (see Figure 3 for distributions) with a sharp drop at the 98th percentile. These values then can reach up to hundreds or thousands of REUs. As these values are not reproducible across multiple runs of the FastRelax protocol across different seeds, we assume that they occur due to FastRelax failing to find a local minimum. Thus, we discard sequences with values below 0 in the training dataset and during evaluation to reduce noise. To verify that this filtering choice does not drive our conclusions, we additionally compute every Rosetta result table without the threshold (i.e. retaining all FastRelax outliers) and report the unfiltered tables in Section A.9 (Table 5 for the first antigen and Table 6 for the second).

In the Rosetta task, we curated two sequence distributions with respective energy values. In the random dataset, we used 2448 random unique CDR sequences and respective affinity measures. For the expert dataset, we recreated the computational sequence design pipeline as utilized by Dauparas et al. (2022) and Vázquez Torres et al. (2025), which was used to generate sequences successfully verified in multiple real-world experiments. In this approach, we first sampled 1000 CDR backbone structures using RFdiffusion (Watson et al., 2023) conditioned on the nanobody and antigen structure. We then applied Protein-MPNN (Dauparas et al., 2022) to sample a total of 2483 unique sequences likely to fold into the respective structures. We evaluate the generated sequences using the same steps outlined above to prevent mismatches between the training and evaluation analyses.

## A.2 Tabular Results

## A.3 Absolut!

In Table 1 we show evaluation metrics of our method and all baselines on three data distributions in the Absolut! environment.

| | Method | Absolut! Expert | Absolut! Natural | Absolut! Random |
|---|---|---|---|---|
| Free Energy & Top 100 | Dataset | -110.53 ± 0.00 | -116.46 ± 0.00 | -86.21 ± 0.00 |
| | SeqDiff | -104.95 ± 1.62 / -123.51 ± 1.03 | -109.48 ± 1.44 / -118.69 ± 0.47 | -86.49 ± 0.72 / -99.03 ± 0.87 |
| | SeqDiff+QL | **-120.02 ± 3.16** / **-129.37 ± 0.78** | -110.00 ± 1.13 / **-122.73 ± 0.77** | -95.80 ± 2.13 / -107.48 ± 1.67 |
| | SimDiff | -111.30 ± 1.18 / -126.48 ± 0.58 | -110.00 ± 1.71 / -119.68 ± 0.77 | -86.76 ± 2.42 / -98.99 ± 2.85 |
| | GFlowNet | -103.87 ± 0.41 / -117.94 ± 0.73 | -108.25 ± 0.42 / -114.25 ± 0.51 | **-104.62 ± 0.28** / **-111.05 ± 0.56** |
| | BC | -110.02 ± 0.99 / -116.20 ± 3.59 | **-113.73 ± 0.49** / -114.59 ± 1.04 | -86.23 ± 0.71 / -89.36 ± 2.13 |
| | BC+QL | **-119.58 ± 0.45** / -127.82 ± 0.15 | **-113.80 ± 0.96** / -118.33 ± 1.96 | -93.46 ± 0.25 / -104.64 ± 0.70 |
| Diversity & Novelty | Dataset | 7.72 ± 0.00 | 7.38 ± 0.00 | 10.27 ± 0.00 |
| | SeqDiff | 8.08 / 3.57 | 8.00 / 2.63 | 10.25 / 6.48 |
| | SeqDiff+QL | 5.18 / 2.23 | 6.97 / 2.58 | 6.89 / 6.17 |
| | SimDiff | 7.28 / 3.13 | 7.51 / 2.47 | 10.15 / 6.57 |
| | GFlowNet | 9.32 / 6.32 | 6.05 / 5.72 | 8.28 / 6.64 |
| | BC | 7.78 / 2.54 | 7.59 / 1.77 | 10.26 / 5.91 |
| | BC+QL | 6.46 / 1.85 | 6.97 / 1.60 | 9.88 / 6.01 |
| Novel & Duplicated | SeqDiff | 500 ± 0 / 959 ± 65 | 500 ± 0 / 870 ± 87 | 500 ± 0 / 318 ± 25 |
| | SeqDiff+QL | 500 ± 0 / 2049 ± 1579 | 500 ± 0 / 1585 ± 706 | 500 ± 0 / 214 ± 153 |
| | SimDiff | 500 ± 0 / 9 ± 4 | 500 ± 0 / 48 ± 8 | 500 ± 0 / 0 ± 0 |
| | GFlowNet | 500 ± 0 / 0 ± 0 | 500 ± 0 / 0 ± 0 | 500 ± 0 / 0 ± 0 |
| | BC | 141 ± 24 / 5594 ± 538 | 104 ± 6 / 5016 ± 6 | 130 ± 19 / 5399 ± 542 |
| | BC+QL | 352 ± 22 / 5792 ± 22 | 158 ± 26 / 5782 ± 433 | 366 ± 27 / 5778 ± 27 |

Table 1: Summary of results across different methods and metrics on the Absolut! task. Best results are underlined, results that are not significantly worse than the best are bold.

### A.3.1 Rosetta

In Table 2 we show evaluation metrics of our method and all baselines on two data distributions in the Rosetta environment.

| | Method | Rosetta Expert | Rosetta Random |
|---|---|---|---|
| **Free Energy & Top 100** | Dataset | -32.75 ± 0.00 | -27.67 ± 0.00 |
| | SeqDiff | -33.12 ± 0.55 / -38.49 ± 0.30 | -26.64 ± 0.39 / -34.54 ± 0.43 |
| | SeqDiff+QL | **-37.10 ± 0.79** / **-41.47 ± 0.53** | -27.28 ± 0.76 / -35.23 ± 0.70 |
| | SimDiff | -34.75 ± 0.69 / -40.47 ± 0.68 | -26.98 ± 1.31 / -34.85 ± 1.57 |
| | GFlowNet | -29.64 ± 0.31 / -37.14 ± 0.32 | **-30.88 ± 1.36** / **-38.81 ± 1.51** |
| | BC | -32.46 ± 0.46 / -37.84 ± 0.42 | -26.72 ± 0.44 / -34.79 ± 0.58 |
| | BC+QL | -35.03 ± 0.16 / -39.76 ± 0.45 | -28.06 ± 0.48 / -36.46 ± 0.43 |
| **Diversity & Novelty** | Dataset | 18.05 ± 0.00 | 25.49 ± 0.00 |
| | SeqDiff | 18.22 / 9.38 | 25.46 / 20.10 |
| | SeqDiff+QL | 15.16 / 8.94 | 22.66 / 20.10 |
| | SimDiff | 18.19 / 11.21 | 24.81 / 20.20 |
| | GFlowNet | 22.37 / 18.37 | 20.09 / 20.46 |
| | BC | 17.98 / 7.11 | 25.54 / 18.91 |
| | BC+QL | 17.42 / 7.55 | 24.84 / 19.17 |
| **Novel & Duplicated** | SeqDiff | 497 ± 0 / 0 ± 0 | 487 ± 5 / 0 ± 0 |
| | SeqDiff+QL | 499 ± 1 / 0 ± 0 | 487 ± 7 / 0 ± 0 |
| | SimDiff | 497 ± 3 / 0 ± 0 | 490 ± 4 / 0 ± 0 |
| | GFlowNet | 498 ± 2 / 0 ± 0 | 498 ± 2 / 0 ± 0 |
| | BC | 498 ± 1 / 1032 ± 75 | 493 ± 3 / 3098 ± 905 |
| | BC+QL | 498 ± 1 / 3866 ± 205 | 485 ± 17 / 3787 ± 1303 |

Table 2: Summary of results across different methods and metrics on the Rosetta task. Best results are underlined, results that are not significantly worse than the best are bold.

### A.4 Implementation Details

### A.4.1 Hyperparameter search

For all methods we implemented and task configurations we tested the following hyperparameter settings, given that the hyperparameter was applicable:

- $\eta \in [0.1, 0.3, 0.5, 0.7, 0.9, 0.95]$

- training duration $\in [50, 100, 150, 200]$

We found that for the narrow natural dataset on Absolut! a higher focus on BC was necessary to prevent distribution shift.

We then selected the following hyperparameters: Training episodes (with 1000 gradient updates each):

- 200 for all Absolut! tasks

- 50 for Rosetta expert task, except for BC+QL, where 200 training epochs let to better results

- 200 for Rosetta random task

$\eta$ was used for SeqDiff+QL and BC+QL:

- Absolut! expert 0.95 for both

- Absolut! natural 0.5 for both

- Absolut! random 0.95 for SeqDiff+QL and 0.9 BC+QL

- Rosetta expert 0.9 for SeqDiff+QL and 0.95 BC+QL

- Rosetta random 0.9 for SeqDiff+QL and 0.95 BC+QL

For the GFlowNet implementation, we utilized the original code released by Jain et al. (2022) and did a hyperparameter search over training duration and reward exponent ($\beta$ in the respective publication). For the reward exponent, we tested $\beta \in [3, 5, 10, 25, 50]$ and found:

- $\beta = 3$ to work best on Absolut! expert

- $\beta = 3$ to work best on Absolut! natural

- $\beta = 5$ to work best on Absolut! random

- $\beta = 10$ to work best on both Rosetta tasks

and a training duration of 50 epochs to be best for all tasks except Absolut! random, where 150 epochs led to slightly better results. Note that a higher $\beta$ for GFlowNet can slightly reduce the mean free energy, but leads to significantly increased duplicates and diversity loss, as the policies start overfitting to model-bias (e.g., only generating sequences with high counts of AA F.

### A.4.2 Additional Details

To reduce the effect of the latent space's structure on the reported results, we share the pre-trained VAE between all datasets for a given seed. Due to the large computational burden, we chose $N = 10$ diffusion steps for our experiments. We follow (Wang et al., 2023) for the choice of $\beta$ noise schedule to train our diffusion model.

In contrast to the implementation by Wang et al. (2023), we do not generate 50 actions using the Diffusion Model per step and sample the final action via a softmax distribution over the respective Q-weights. Instead, we directly take the action sampled from the diffusion model.

When conditioning a policy $\pi$ on a state $s_t$, we transform the state into a token-representation using an embedding layer instead of the learned VAE representation we use for AAs.

For all tested methods, normalize all rewards in the training distribution between 0 and 1 to prevent reward shifts between datasets and evaluation schemes.

### A.5 Baseline Algorithms

### A.5.1 BC and BC+QL

Inspired by prior work (Fujimoto & Gu, 2021; Nair et al., 2021; Goecks et al., 2019), we create a stochastic actor-critic agent balancing behavior cloning and QL in a categorical action setting. This combination, which we refer to as BC+QL, was previously successfully applied in continuous action settings and showed remarkable performance in the offline RL setting. The actor is implemented using a simple feedforward network $\pi_\theta$ predicting a probability vector $\boldsymbol{p}$ containing all 20 possible next AAs given the current sequence as $\boldsymbol{p}_t = \pi_\theta(s_t)$. Thus, we iteratively sample from the stochastic policy to generate entire sequences. The behavior cloning part is trained using cross-entropy loss to model the dataset distribution $L_{BC} = \mathbb{E}_{(s_t,a_t)\sim D, \boldsymbol{p}=\pi_\theta(s_t)}[-\sum_{k=1}^{20} y_k \log \boldsymbol{p}_k, y_k = 1[k = a_t]]$. This loss alone is used to create the BC agent.

For policy optimization, we add a Q-function $Q_\phi(s_t)$ estimation the action values for all 20 AAs given the incomplete sequence $s_t$ and optimize $Q_{\phi_i}$ to minimize $L_{Q_i} = \mathbb{E}_{(s_t,a_t,s_{t+1})\sim D}[||R(s_t,a_t) + \min_{j=1,2}(Q_{\phi'_j}(s_{t+1})\pi'_\theta(s_{t+1})) - Q_{\phi_i}(s_t,a_t)||^2]$, and policy $\pi_\theta$ to minimize $(1-\eta)L_{BC}(\theta) - \eta \mathbb{E}_{s\sim D}[Q_\phi(s)\pi_\theta(s_t)]$, where $\eta$ can be tuned to balance the loss terms. This algorithm, referred to as BC+QL, despite its simplicity, fulfills the requirements of being non-deterministic, suited for offline learning, and capable of objective-driven design, thus serving as a minimalist and lightweight baseline.

### A.5.2 SimDiff

Alternatively to our chosen incremental approach to sequence design, the entire sequence could be designed at once using a diffusion model. We utilize a DDPM (Ho et al., 2020) for this purpose and refer to the agent as SimDiff. Specifically, we use latent diffusion by representing all sequences of length $L$ in the training dataset in a concatenation of their amino acid's 2D latent representations using the pretrained VAE, $z_{cat}$. $z_{cat} = \text{concat}(z_0, z_1, \ldots, z_{L-1})$, $z_l = e_\omega(a)$, where $e_\omega$ is trained as described in Section 4.1. After sampling, we decode the discrete AAs using the VAE. While this baseline is not capable of objective-driven design, it can be used to estimate a potential performance gap between unguided incremental and unguided simultaneous diffusion.

### A.5.3 SeqDiff

In SeqDiff+QL we employ a BC loss to constrain the agent to stay close to the training data distribution. If we only keep this loss, by setting $\eta = 0$, we create an algorithm, SeqDiff, which models the training data distribution by sequentially sampling AAs. This algorithm is suited for diverse generation and offline training, but not for objective-driven design. Using this baseline, we can analyze a potential gap between sequential and simultaneous generation using diffusion models.

### A.5.4 GFlowNet

Introduced by Jain et al. (2022) for biological sequence design, this algorithm uses Generative Flow Networks (Bengio et al., 2021) in combination with a learned proxy reward model. Specifically, the proxy reward model is implemented as an ensemble with mean $\mu$ and an uncertainty as the standard deviation $\sigma$ of the ensemble. The model is trained to maximize the upper confidence bound as $\mu + 0.1\sigma$.

### A.6 Amino Acid Groups

Our grouping of AAs is taken from the classification by Garrett & Grisham (2010) and given as follows:

- Nonpolar (hydrophobic) AAs: L, P, A, V, M, W, F, I
- Polar, uncharged AAs: G, S, N, Q, T, C, Y
- Acidic AAs: D, E
- Basic AAs: K, R, H

Note that we chose this specific grouping not because we are convinced it bears an advantage, but rather because it was the most prominent grouping we found in the literature.

### A.7 Bias in evaluation Environments

To examine why our induced priors have a positive effect on all Absolut! environments, but not the Rosetta environments, and why diversity of sequences in the Absolut! environment decreases with increased affinity, we analyzed the shift in AA distribution between all sequences in the random datasets and the top 5% sequences in the random dataset. In the Absolut! dataset, the fraction of hydrophobic AAs (colored blue in Figure 6) increased from 39.77% to 55.63%, indicating a bias towards such AAs. This is further emphasized by the natural dataset, which represents the top 0.01% to 0.1% of 6.9 million tested murine CDRH3 sequences in the Absolut! publication Robert et al. (2022). Here, hydrophobic AAs make up 60.31% of the dataset. In the Rosetta dataset, while we observe that certain positions favor specific AA groups, we observe no general bias towards a single AA group. However, we can observe that in the Rosetta task, position 9 seems to favor polar, uncharged AAs and position 14 seems to favor hydrophobic AAs, indicating a relevance for binding the antigen. This shows that the assumed prior was thus only present in one of our environments.

To corroborate this analysis directly at the diversity level, we additionally analyze the 6.9 million murine CDRH3 sequences scored by Robert et al. (2022). This yields 4,316,741 unique murine sequence-energy

pairs after deduplication. We then bin the energies into 100 equal-width bins and compute the diversity over up to 1,000 randomly sampled sequences per bin. The resulting empirical curve, shown in Figure 7, exhibits a decrease in diversity as predicted energy decreases (binding strengthens) over the energy range. Note that at the edges of the energy range, the empirical reference becomes unreliable: bins below $-121$ and above $-72$ contain only single- to low-double-digit numbers of sequences, and the apparent uptick in diversity at the very low- and very high-energy ends is therefore driven by sampling noise rather than by a real reversal of the trend. For the model-generated sequences, we take all generated sequences across all methods trained on the Absolut! environment and antigen 2DD8, which show the same trend of decreasing diversity with decreasing energy. Note that neither of the two data distributions is unbiased, as the murine distribution is biased towards sequences that are more likely to occur in nature, while the model-generated distribution is biased towards sequences that follow a trend in the training dataset. Thus none of them are suited for a faithful estimation of the true diversity-energy relationship, which would require an unbiased sampling of the entire sequence space, which is infeasible.

## A.8 Second-antigen evaluation

To probe whether our findings carry across antigens, we evaluate every method on a second antigen per oracle. For Absolut! we use human CD38 (PDB: 3RAJ); for Rosetta we use a nanobody binding PaaR2 (PDB: 8C3K). Training dataset semantics, hyperparameters, and the five seeds match those used for the first antigen. The corresponding free-energy distributions and auxiliary metrics are shown in Figure 8 (Absolut!) and Figure 9 (Rosetta), with full tabular results in Table 3 and Table 4.

For 8C3K we observe a small fraction of sequences with extremely low estimated energies (below $-60$ REUs), analogous to the high-energy FastRelax outliers discussed in Section A.1. To preserve readability of the violin and parallel-coordinate plots in Figure 9, sequences with energy below $-60$ REUs are clipped from the visualization only; they are retained in all reported energy statistics and tabular results. This concerns 0.175% of evaluated sequences in the expert 8C3K setting.

| | Method | Absolut! Expert | Absolut! Natural | Absolut! Random |
|---|---|---|---|---|
| **Free Energy & Top 100** | Dataset | $-98.39 \pm 0.00$ | $-106.09 \pm 0.00$ | $-81.90 \pm 0.00$ |
| | SeqDiff | $-95.32 \pm 1.93$ / $-111.41 \pm 2.14$ | $-97.49 \pm 0.50$ / $-108.02 \pm 0.42$ | $-82.10 \pm 0.84$ / $-93.85 \pm 1.06$ |
| | SeqDiff+QL | **$-108.64 \pm 3.23$** / **$-119.71 \pm 0.82$** | $-99.18 \pm 1.85$ / $-114.85 \pm 1.27$ | $-88.03 \pm 1.52$ / $-99.28 \pm 1.76$ |
| | SimDiff | $-99.15 \pm 1.67$ / $-113.77 \pm 1.20$ | $-95.60 \pm 0.67$ / $-108.25 \pm 0.50$ | $-82.71 \pm 2.30$ / $-94.37 \pm 2.48$ |
| | GFlowNet | $-93.67 \pm 0.38$ / $-108.69 \pm 1.24$ | **$-104.60 \pm 2.17$** / **$-124.28 \pm 0.99$** | **$-96.98 \pm 0.96$** / **$-114.97 \pm 1.47$** |
| | BC | $-98.88 \pm 1.24$ / $-107.67 \pm 1.31$ | $-100.91 \pm 0.07$ / $-108.57 \pm 0.22$ | $-82.07 \pm 0.42$ / $-86.61 \pm 0.74$ |
| | BC+QL | **$-106.63 \pm 0.52$** / $-118.24 \pm 0.38$ | $-101.66 \pm 0.66$ / $-110.36 \pm 0.38$ | $-85.97 \pm 0.55$ / $-97.08 \pm 0.67$ |
| **Diversity & Novelty** | Dataset | $8.30 \pm 0.00$ | $8.06 \pm 0.00$ | $10.27 \pm 0.00$ |
| | SeqDiff | 8.43 / 3.75 | 8.41 / 2.88 | 10.25 / 6.47 |
| | SeqDiff+QL | 5.33 / 2.29 | 6.22 / 3.05 | 7.08 / 6.14 |
| | SimDiff | 8.01 / 3.69 | 8.11 / 3.03 | 10.16 / 6.55 |
| | GFlowNet | 8.53 / 5.92 | 4.28 / 4.32 | 8.27 / 6.63 |
| | BC | 8.32 / 2.83 | 8.10 / 2.06 | 10.25 / 5.92 |
| | BC+QL | 7.02 / 1.96 | 7.85 / 2.10 | 10.10 / 6.00 |
| **Novel & Duplicated** | SeqDiff | $500 \pm 0$ / $972 \pm 76$ | $500 \pm 0$ / $193 \pm 26$ | $500 \pm 0$ / $330 \pm 26$ |
| | SeqDiff+QL | $469 \pm 69$ / $2600 \pm 1985$ | $500 \pm 0$ / $306 \pm 101$ | $500 \pm 0$ / $172 \pm 204$ |
| | SimDiff | $500 \pm 0$ / $4 \pm 3$ | $500 \pm 0$ / $6 \pm 3$ | $500 \pm 0$ / $0 \pm 0$ |
| | GFlowNet | $500 \pm 0$ / $0 \pm 0$ | $500 \pm 0$ / $51 \pm 30$ | $500 \pm 0$ / $0 \pm 0$ |
| | BC | $173 \pm 16$ / $5971 \pm 16$ | $459 \pm 20$ / $5685 \pm 20$ | $153 \pm 8$ / $5991 \pm 8$ |
| | BC+QL | $437 \pm 10$ / $5707 \pm 10$ | $371 \pm 19$ / $5773 \pm 19$ | $424 \pm 48$ / $5720 \pm 48$ |

Table 3: Summary of Absolut! 3RAJ results. Best results are underlined; results that are not significantly worse than the best are in bold.

A few observations stand out. On the second Absolut! antigen the qualitative ranking is preserved on expert (SeqDiff+QL best) and random (GFlowNet best); on natural the ordering inverts in favor of GFlowNet, which we attribute, as discussed in Section 5.4.4, to the natural high-affinity bias of the natural dataset on this antigen aligning particularly well with the GFlowNet proxy. This is further supported by the logo plots in Figure 10, which show that GFlowNet generations on the second Absolut! antigen collapse onto a small set of hydrophobic residues, illustrating that the policy exploits a narrow region of sequence space favored by the learned proxy. On the second Rosetta antigen, the expert dataset is itself already very high-

| | Method | Rosetta Expert | Rosetta Random |
|---|---|---|---|
| Free Energy & Top 100 | Dataset | -40.53 ± 0.00 | -20.02 ± 0.00 |
| | SeqDiff | **-41.06 ± 0.39** / **-45.31 ± 0.15** | -19.96 ± 0.48 / -32.08 ± 1.06 |
| | SeqDiff+QL | **-38.85 ± 4.64** / **-45.84 ± 1.24** | -22.09 ± 2.26 / -36.58 ± 4.85 |
| | SimDiff | **-41.31 ± 0.33** / **-45.41 ± 0.09** | -20.14 ± 1.36 / -32.06 ± 2.62 |
| | GFlowNet | -29.50 ± 1.80 / -39.27 ± 0.52 | -23.71 ± 0.95 / -33.91 ± 0.95 |
| | BC | -40.45 ± 0.37 / **-45.29 ± 0.17** | -20.08 ± 0.14 / -32.61 ± 0.24 |
| | BC+QL | **-25.77 ± 14.10** / **-34.82 ± 10.43** | **-35.43 ± 0.81** / **-43.99 ± 0.39** |
| Diversity & Novelty | Dataset | 8.44 ± 0.00 | 18.39 ± 0.00 |
| | SeqDiff | 8.32 / 2.09 | 18.36 / 12.19 |
| | SeqDiff+QL | 8.97 / 4.98 | 15.49 / 13.13 |
| | SimDiff | 8.08 / 2.75 | 17.98 / 13.73 |
| | GFlowNet | 12.48 / 12.70 | 16.64 / 13.76 |
| | BC | 8.45 / 0.29 | 18.39 / 1.53 |
| | BC+QL | 8.35 / 9.27 | 17.70 / 10.27 |
| Novel & Dupli-cated | SeqDiff | 496 ± 2 / 7 ± 3 | 482 ± 5 / 2 ± 1 |
| | SeqDiff+QL | 446 ± 64 / 36 ± 17 | 488 ± 3 / 66 ± 116 |
| | SimDiff | 498 ± 1 / 1 ± 1 | 475 ± 19 / 0 ± 0 |
| | GFlowNet | 497 ± 2 / 0 ± 0 | 496 ± 2 / 0 ± 0 |
| | BC | 499 ± 1 / 64 ± 11 | 497 ± 5 / 64 ± 19 |
| | BC+QL | 293 ± 131 / 3191 ± 2299 | 492 ± 3 / 2709 ± 425 |

Table 4: Summary of Rosetta 8C3K results. Best results are underlined; results that are not significantly worse than the best are in bold.

affinity, so the gains from Q-guidance over unguided diffusion baselines shrink and several baselines become statistically tied with SeqDiff+QL on top-100 affinity; on random the lightweight BC+QL baseline performs best. The first- and second-antigen results together suggest a consistent picture: Q-guidance provides clear benefits whenever there is meaningful affinity headroom in the dataset, while the relative ranking among objective-driven methods can shift across antigens.

### A.9 Sensitivity to the Rosetta energy threshold

The main Rosetta results in Section 5.4.2 drop sequences whose FastRelax energy lies above zero (see Section A.1 for the rationale). The two tables below repeat all Rosetta result computations, including the full main-text table from Table 2 and the second-antigen table from Table 4, without that filter, so that every FastRelax outlier is included. Comparing each row to its filtered counterpart confirms that the filter shifts mean energies by only a small margin and never reverses a method comparison; we therefore retain the filter in the main text on noise-reduction grounds while reporting the unfiltered numbers here for transparency.

### A.10 Distribution-shift analysis

To make the trust-region behavior of the BC term explicit, we sweep the Q-weight $\eta$ on Absolut! expert and visualize the resulting amino-acid latent-space distributions of generated actions against the training set in Figure 11, alongside the corresponding sequence logo plots at $\eta \in \{0.0, 0.5, 0.97\}$.

The two views agree: as $\eta$ increases, mass in latent space migrates from the training-data density towards a small number of high-Q latents, and the position-wise sequence composition shifts away from the training distribution accordingly. This makes the distribution-shift behavior controlled by $\eta$ explicit and quantitative.

### A.11 Latent dimensionality ablation

We ablate the per-token VAE latent dimensionality $d \in \{1, 2, 3, 4, 5\}$ on Absolut! expert, with all other hyperparameters fixed at the main-text values (Figure 13). We find a clear tradeoff: $d = 1$ underperforms even after extended VAE training, whereas $d \geq 3$ reaches slightly better predicted affinity at the cost of markedly lower novelty and diversity. We attribute this to the additional latent volume per amino acid, which eases fitting the training data with a diffusion policy but also promotes overfitting. We therefore adopt $d = 2$ as a favorable point on this curve, and treat the dimensionality as an application-level choice rather than a fundamental constraint.

| | Method | Rosetta Expert | Rosetta Random |
|---|---|---|---|
| **Free Energy & Top 100** | Dataset | -32.33 ± 0.00 | -25.44 ± 0.00 |
| | SeqDiff | -32.85 ± 0.58 / -38.49 ± 0.30 | -24.04 ± 1.00 / -34.54 ± 0.43 |
| | SeqDiff+QL | **-36.94 ± 1.02** / **-41.47 ± 0.53** | -24.98 ± 1.72 / -35.23 ± 0.70 |
| | SimDiff | -34.42 ± 0.86 / -40.47 ± 0.68 | -25.10 ± 1.64 / -34.85 ± 1.57 |
| | GFlowNet | -29.31 ± 0.20 / -37.14 ± 0.32 | **-30.55 ± 1.19** / **-38.81 ± 1.51** |
| | BC | -32.22 ± 0.51 / -37.84 ± 0.42 | -25.29 ± 1.11 / -34.79 ± 0.58 |
| | BC+QL | -34.83 ± 0.06 / -39.76 ± 0.45 | -26.56 ± 0.62 / -36.46 ± 0.43 |
| **Diversity & Novelty** | Dataset | 18.06 ± 0.00 | 25.49 ± 0.00 |
| | SeqDiff | 18.23 / 9.38 | 25.46 / 20.09 |
| | SeqDiff+QL | 15.17 / 8.95 | 22.67 / 20.10 |
| | SimDiff | 18.19 / 11.20 | 24.81 / 20.19 |
| | GFlowNet | 22.37 / 18.37 | 20.11 / 20.45 |
| | BC | 17.98 / 7.12 | 25.54 / 18.90 |
| | BC+QL | 17.42 / 7.55 | 24.85 / 19.18 |
| **Novel & Duplicated** | SeqDiff | 500 ± 0 / 0 ± 0 | 500 ± 0 / 0 ± 0 |
| | SeqDiff+QL | 500 ± 0 / 0 ± 0 | 500 ± 0 / 0 ± 0 |
| | SimDiff | 500 ± 0 / 0 ± 0 | 500 ± 0 / 0 ± 0 |
| | GFlowNet | 500 ± 0 / 0 ± 0 | 500 ± 0 / 0 ± 0 |
| | BC | 500 ± 0 / 1032 ± 75 | 500 ± 0 / 3098 ± 905 |
| | BC+QL | 500 ± 0 / 3866 ± 205 | 493 ± 16 / 3787 ± 1303 |

Table 5: Rosetta first-antigen (8HXQ) results without the energy-threshold filter (corresponds to Table 2).

| | Method | Rosetta Expert | Rosetta Random |
|---|---|---|---|
| **Free Energy & Top 100** | Dataset | -39.65 ± 0.00 | -5.39 ± 0.00 |
| | SeqDiff | **-40.56 ± 0.55** / **-45.31 ± 0.15** | -13.86 ± 1.91 / -32.08 ± 1.06 |
| | SeqDiff+QL | **-30.09 ± 14.36** / **-45.84 ± 1.24** | -18.78 ± 3.10 / -36.58 ± 4.85 |
| | SimDiff | **-41.08 ± 0.35** / **-45.41 ± 0.09** | -9.89 ± 8.31 / -32.06 ± 2.62 |
| | GFlowNet | -29.04 ± 1.71 / -39.27 ± 0.52 | -22.71 ± 1.27 / -33.91 ± 0.95 |
| | BC | -40.38 ± 0.33 / **-45.29 ± 0.17** | -19.10 ± 1.64 / -32.61 ± 0.24 |
| | BC+QL | -11.08 ± 21.16 / **-34.82 ± 10.43** | **-33.93 ± 1.39** / **-43.99 ± 0.39** |
| **Diversity & Novelty** | Dataset | 8.51 ± 0.00 | 18.40 ± 0.00 |
| | SeqDiff | 8.35 / 2.10 | 18.36 / 12.21 |
| | SeqDiff+QL | 9.05 / 5.05 | 15.51 / 13.11 |
| | SimDiff | 8.09 / 2.76 | 17.98 / 13.70 |
| | GFlowNet | 12.48 / 12.70 | 16.64 / 13.74 |
| | BC | 8.46 / 0.29 | 18.39 / 1.57 |
| | BC+QL | 8.27 / 9.36 | 17.71 / 10.29 |
| **Novel & Duplicated** | SeqDiff | 500 ± 0 / 7 ± 3 | 500 ± 0 / 2 ± 1 |
| | SeqDiff+QL | 500 ± 0 / 36 ± 17 | 500 ± 0 / 66 ± 116 |
| | SimDiff | 500 ± 0 / 1 ± 1 | 500 ± 0 / 0 ± 0 |
| | GFlowNet | 500 ± 0 / 0 ± 0 | 500 ± 0 / 0 ± 0 |
| | BC | 500 ± 0 / 64 ± 11 | 500 ± 0 / 64 ± 19 |
| | BC+QL | 401 ± 165 / 3191 ± 2299 | 500 ± 0 / 2709 ± 425 |

Table 6: Rosetta second-antigen (8C3K) results without the energy-threshold filter (corresponds to Table 4).

We further note that the practical alternative to a small continuous latent is not a discrete representation: abandoning continuity (e.g., one-hot or learned discrete tokens) would forfeit the gradient-based Q-guidance that motivates the diffusion-policy formulation. The relevant alternative is therefore a higher-dimensional continuous latent, at the cost of less novel generations.

## A.12   Q-value calibration

Table 7 reports the Spearman rank correlation between learned Q-values and the actual oracle energy of generated sequences, separately for both antigens, all dataset distributions, and the Q-using methods. Values are mean ± std across seeds. Note that the correlation is computed over novel generated sequences only, so it reflects the Q-function's ability to generalize beyond the training set. Furthermore, a policy might still generate high-affinity sequences even with a low Spearman correlation if the Q-function successfully discriminates low-affinity sequences despite having a low correlation in the high-affinity regime. In such a case, however, the filtering method is unlikely to perform well.

| Dataset | 1st antigen: 2DD8 and 8HXQ | | | | |
| --- | --- | --- | --- | --- | --- |
| | Absolut! Expert | Absolut! Natural | Absolut! Random | Rosetta Expert | Rosetta Random |
| BC+QL | $0.860 \pm 0.021$ | $0.468 \pm 0.109$ | $0.478 \pm 0.025$ | $0.757 \pm 0.043$ | $0.487 \pm 0.071$ |
| GFlowNet | $-0.243 \pm 0.067$ | $0.369 \pm 0.024$ | $0.612 \pm 0.038$ | $0.644 \pm 0.035$ | $0.577 \pm 0.063$ |
| SeqDiff+QL | $0.587 \pm 0.152$ | $0.224 \pm 0.030$ | $0.207 \pm 0.049$ | $0.541 \pm 0.037$ | $0.159 \pm 0.089$ |
| Dataset | 2nd antigen: 3RAJ and 8C3K | | | | |
| | Absolut! Expert | Absolut! Natural | Absolut! Random | Rosetta Expert | Rosetta Random |
| BC+QL | $0.808 \pm 0.017$ | $0.358 \pm 0.067$ | $0.431 \pm 0.065$ | $0.145 \pm 0.352$ | $0.599 \pm 0.064$ |
| GFlowNet | $0.224 \pm 0.063$ | $0.569 \pm 0.049$ | $0.269 \pm 0.059$ | $0.531 \pm 0.084$ | $0.707 \pm 0.024$ |
| SeqDiff+QL | $0.594 \pm 0.125$ | $0.176 \pm 0.046$ | $0.247 \pm 0.111$ | $0.186 \pm 0.217$ | $0.304 \pm 0.057$ |

Table 7: Spearman rank correlation between learned Q-values and oracle energy on the generated sequences (mean $\pm$ std across seeds). SeqDiff+QL is calibrated on every expert setting and degrades on random Rosetta, which is precisely the setting on which Q-filtering fails in Figure 5(b).

## A.13  CDR-length scalability

The longest design task in our benchmark is the $L = 28$ Rosetta task that re-designs all three CDRs of a single-domain antibody. For nanobody design specifically, we deliberately restrict to the CDRs because the framework regions show very limited natural sequence diversity and therefore little design headroom. Thus, while it would be feasible to also design these, an agent would, given the training data, presumably overfit to the natural framework sequences for those regions. Extending the framework to (i) full variable domains, (ii) paired heavy/light antibody chains, or (iii) longer CDRH3 loops than 28 residues are natural next steps but lie outside the scope of this paper.

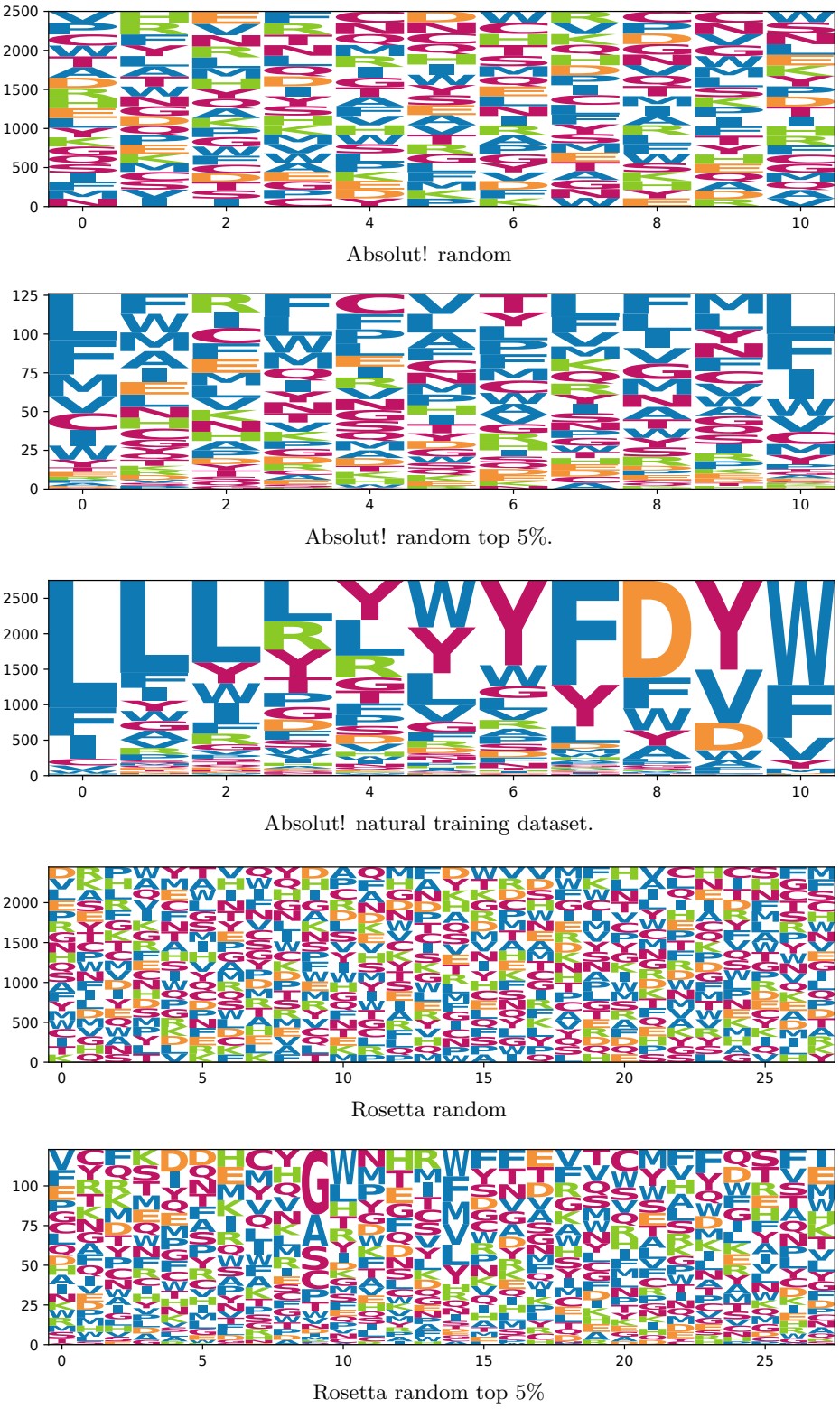

Figure 6: Logo plots for different datasets. Higher affinity datasets in the Absolut! task exhibit a significantly higher fraction of hydrophobic AAs. In the Rosetta task, such tendencies are less observable, and focus on a few positions (e.g., index 9 and 14).

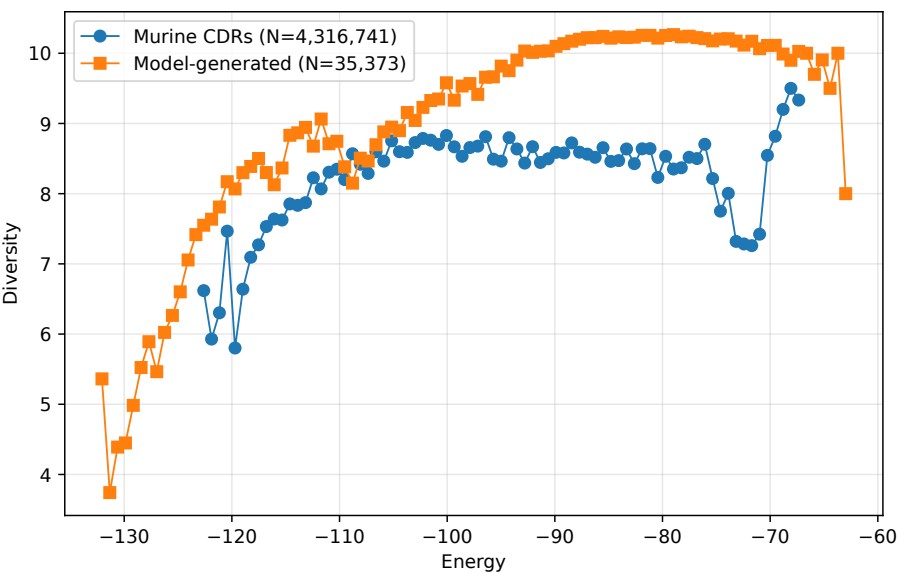

Figure 7: Empirical relationship between predicted free energy and slide-level Levenshtein diversity for antigen 2DD8 in the murine reference set of Robert et al. (2022) and sequences generated by all evaluated methods in our studies. Murine sequence diversity was computed from $N = 4{,}316{,}741$ deduplicated sequence-energy pairs, binned into 100 equal-width energy bins; per bin we draw up to 1,000 sequences uniformly at random and compute the diversity. Diversity decreases as energy decreases (binding strengthens) in the densely-populated central range. Bins at the most negative ($\lesssim -121$) and least negative ($\gtrsim -72$) energies contain only single- to low-double-digit numbers of sequences, so the apparent rise in diversity at the spectrum extremes reflects sampling noise rather than a real reversal of the trend. For model-generated sequences, we took all generated sequences across all methods trained on the Absolut! environment and antigen 2DD8.

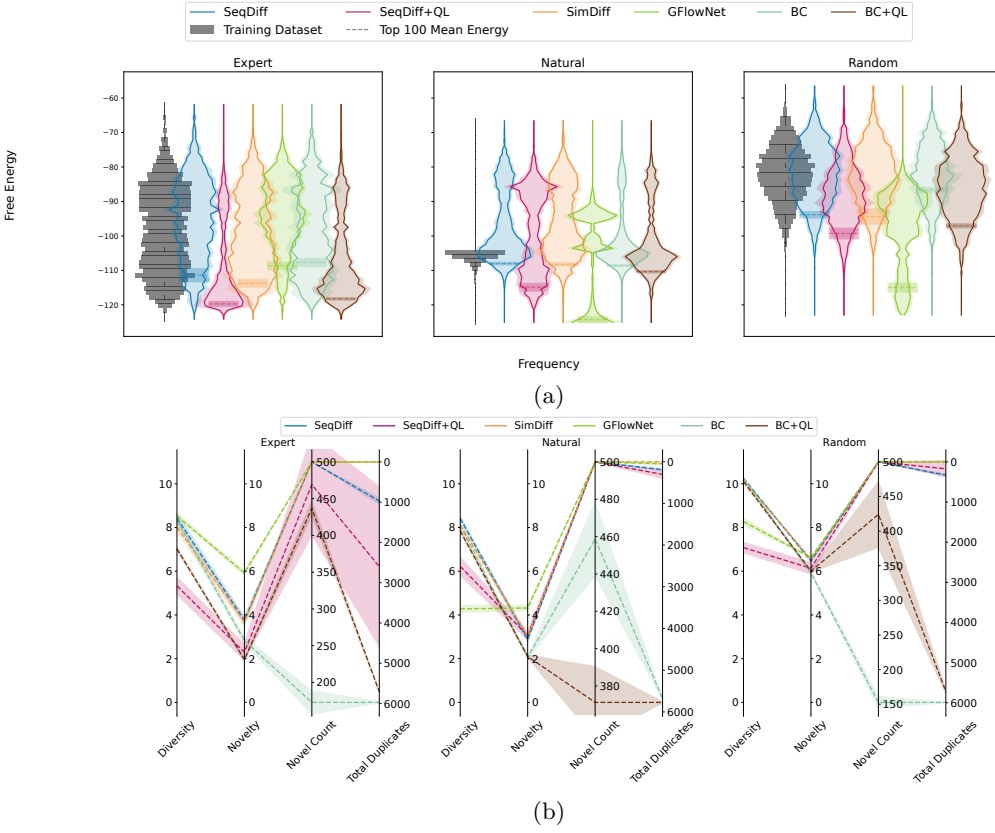

Figure 8: Second-antigen Absolut! results (3RAJ). (a) Free-energy distributions of the top-100 unique novel sequences. (b) Auxiliary metrics (diversity, novelty, novel count, total duplicates).

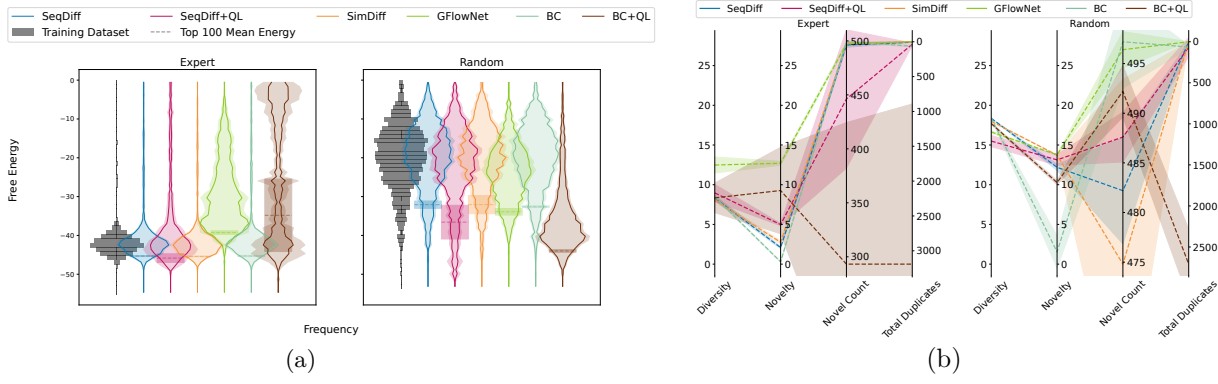

Figure 9: Second-antigen Rosetta results (8C3K). (a) Free-energy distributions of the top-100 unique novel sequences. (b) Auxiliary metrics.

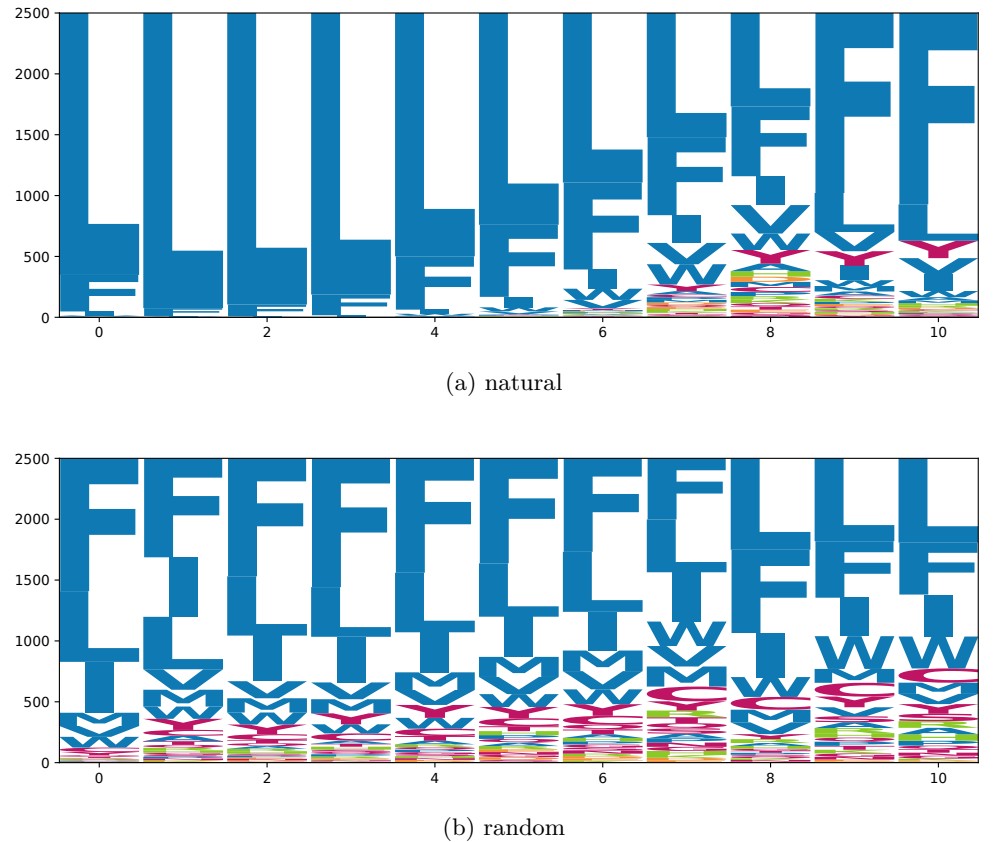

Figure 10: Per-position amino-acid composition of novel sequences produced by GFlowNet on the second Absolut! antigen (3RAJ), for the natural (a) and random (b) datasets. The distribution collapses onto a small set of hydrophobic residues, illustrating that the policy exploits a narrow region of sequence space favored by the learned proxy. This bias correlates with the strong Spearman correlations between proxy and oracle reported in Section A.12 on these datasets and the extreme top-100 free energies seen for GFlowNet in Figure 8.

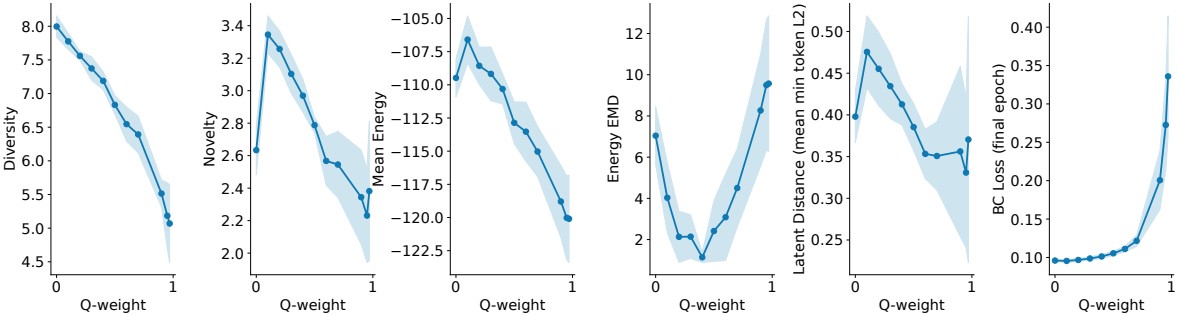

Figure 11: Distribution-shift indicators for SeqDiff+QL on Absolut! expert as a function of the Q-weight $\eta$. From left to right: Diversity, Novelty, mean Energy, Energy EMD against the training-set energy distribution, Latent Distance (mean per-token minimum-L2 distance to the nearest training-set latent), and BC loss at the final training epoch. Lines and shaded bands show mean and standard deviation across seeds. As $\eta$ grows, Diversity and Novelty decrease and the generations shift towards lower-energy (higher-affinity) sequences; the Energy EMD first drops as the generated energies enter the high-affinity tail of the training distribution and then rises once the policy departs from its support. Latent Distance roughly follows Novelty, measuring the same shift in the latent dimension. Lastly, the BC loss exponentially increases with $\eta$.

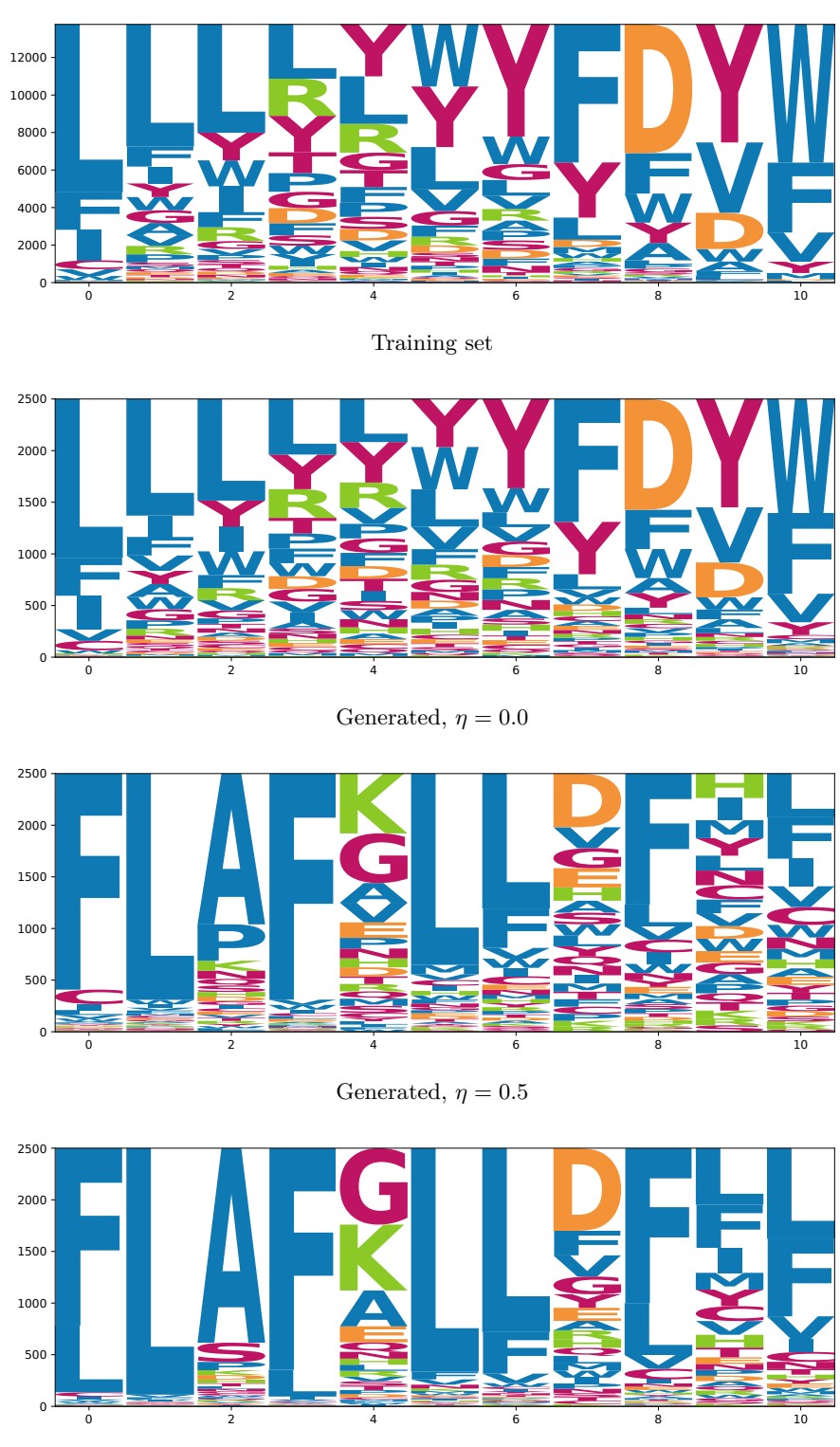

Figure 12: Position-wise amino-acid composition of the training set and of SeqDiff+QL generations under three values of $\eta$ on Absolut! expert. At $\eta = 0.0$ the generated logo closely tracks the training-set logo; at $\eta = 0.5$ a moderate bias towards specific residues appears; at $\eta = 0.97$ the policy concentrates on a small number of high-Q amino acids per position.

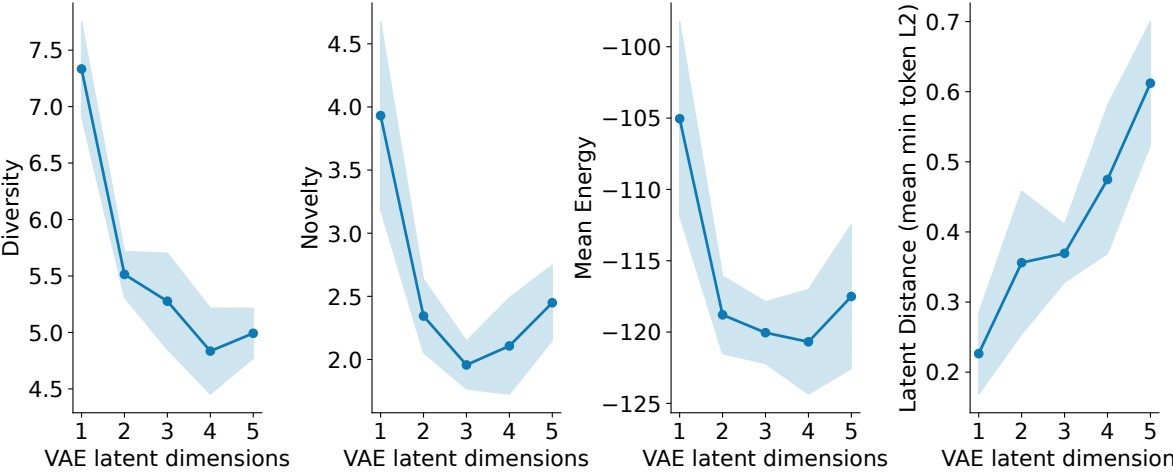

Figure 13: Effect of the per-token VAE latent dimensionality $d \in \{1, 2, 3, 4, 5\}$ on the distribution of generated sequences under SeqDiff+QL on Absolut! expert. Increasing $d$ improves predicted affinity slightly but reduces novelty and diversity.

