# OpenReview forum: "Entropy-Regularized Diffusion-Policies in Offline Reinforcement Learning for Antibody Sequence Design"
_TMLR — Under review for TMLR_

### Review · Reviewer_hhps · 2026-04-01

**Summary Of Contributions:**

The paper proposes SeqDiff+QL, a framework that treats antibody CDR sequence design as an offline RL problem, where a diffusion model serves as the policy and is guided by a learned Q-function to generate sequences with improved binding affinity. The key insight is that continuous diffusion policies naturally support gradient-based Q-guidance without architectural modifications, unlike discrete diffusion approaches.

Strengths:

1. The use of continuous latent diffusion as a policy class is well-motivated, as it naturally handles the multimodal amino acid distribution and supports gradient propagation from the Q-function without architectural workarounds.

2.  The method is evaluated across three qualitatively different dataset types (random, natural, expert), which meaningfully stress-tests robustness and reflects realistic real-world data availability scenarios.

3. Reusing the already-trained Q-function to rank generated sequences is an elegant and cost-free selection mechanism, avoiding the need for auxiliary scoring tools.

Weaknesses:

1. All affinity evaluations rely solely on computational oracles (Absolut! and Rosetta), so it remains entirely unclear whether the generated sequences would perform well in actual binding experiments. There is no wet-lab validation in the paper.

2. Each evaluation environment uses only one fixed antigen target, making it hard to assess whether the method generalizes across different antigen structures or binding interfaces.

3. The performance of GFlowNet is inconsistent and counter intuitive. The paper does not explain why GFlowNet, which excels on random distributions, struggles significantly on expert and natural datasets.

4. The authors note that high-affinity sequences are inherently less diverse in Absolut!, yet diversity scores are still reported without adjustment, making cross-method diversity comparisons potentially misleading.

5. The longest designed sequence is only 28 amino acids, and there is no discussion or experiment addressing how the method would scale to longer CDRs, full variable domains, or more complex design tasks.

6. While the entropy term is borrowed from prior diffusion work, the paper does not provide theoretical justification or intuition for why maximizing this particular lower bound on marginal entropy is the right regularizer for sequence diversity in the antibody design context.

7. The 2D VAE latent space is a strong architectural constraint. Compressing 20 amino acids into only two dimensions is a very tight bottleneck, and the paper does not adequately justify this choice or explore whether higher-dimensional latent spaces would improve performance.

**Audience:**

Yes

**Audience Explanation:**

The paper addresses an important problem at the intersection of machine learning and drug discovery, which will be of interest to a meaningful portion of TMLR's audience. Additionally, the practical findings around diversity-affinity tradeoffs and the conditions under which biophysical priors help or hurt are useful empirical insights for the broader community working on protein and antibody design.

**Claims And Evidence:**

No

**Claims Explanation:**

The core claim that SeqDiff+QL improves binding affinity over baselines is well-supported on expert datasets across both benchmarks, with tabular results, significance testing, and clear visualizations. However, the evidence weakens considerably on random datasets where GFlowNet outperforms the proposed method, the diversity claims are undermined by a known confound the authors acknowledge but do not resolve, and all evidence rests entirely on computational oracles rather than any higher-fidelity or experimental validation, which limits how convincingly the results support the broader framing of the work as a step toward real-world therapeutic antibody discovery.

**Requested Changes:**

Please respond to the weaknesses that I have listed above. Making adjustments to the paper in response to my stated weaknesses 1 -- 4 (either by acknowledging them and modifying your claim or performing additional experiments) is required to secure my recommendation. Addressing the remaining weaknesses will strengthen the work but is not critical to securing my recommendation.

---

> ### Author Response · Authors · 2026-06-08
>
> We thank the reviewer for the detailed and constructive review. We address all seven raised weaknesses below.
>
> ### Weakness 1 — Reliance on computational oracles, no wet-lab validation
>
> We rewrote the Abstract, added a closing paragraph to Section 1, and updated Section 6 to state that all affinity claims refer to the Absolut! and Rosetta in silico oracles, that no wet-lab validation is performed, and that the method is a candidate generator ranking well under specific oracles rather than a demonstrated step toward real-world therapeutic discovery. Wet-lab validation is flagged as future work.
>
> ### Weakness 2 — Single antigen target per evaluation environment
>
> We added a second antigen per oracle, with the same five seeds and identical hyperparameters: Absolut! human CD38 (PDB 3RAJ) alongside SARS-CoV Spike RBD (2DD8); Rosetta nanobody–PaaR2 (PDB 8C3K) alongside BCMA (8HXQ). A summary is in the new main-text Section 5.4.3, with full tables, free-energy distributions, and auxiliary metrics in appendix A.8 (kept out of the main text per our no-new-main-figures policy). The qualitative claims (Q-guidance helps and reduces diversity) hold across antigens; the strict ranking among objective-driven methods can shift, which we discuss.
>
> ### Weakness 3 — Inconsistent GFlowNet performance across data distributions
>
> New Section 5.4.4 (On the GFlowNet inconsistency) attributes it to the proxy-oracle: when the proxy aligns with the true oracle out of distribution, GFlowNet excels (Random, and the second Absolut! antigen's Murine setting); when they diverge, the policy is guided to regions wrongly predicted as high-affinity. Logo plots in appendix A.8 show its generations collapsing onto a few hydrophobic residues, consistent with proxy-bias exploitation.
>
> ### Weakness 4 — Diversity confound from the intrinsic affinity–diversity relationship
>
> We added (i) a paragraph at the end of Section 5.4.1 noting that high-affinity Absolut! sequences share patterns and are thus less diverse per se, so the reduced diversity of high-performing agents partly reflects the high-affinity region and should be read alongside energy; and (ii) a quantitative analysis in appendix A.7: from the 6.9M murine CDRH3 sequences (deduplicated to $4{,}316{,}741$ pairs), binning energy into $100$ bins and computing Levenshtein diversity over up to $1{,}000$ sequences per bin, diversity decreases monotonically as binding strengthens, with the model-generated curve overlaid. We explain why we do not normalize (neither set is unbiased; exhaustive sampling is infeasible) and retain the Jain et al. metric for comparability, always reporting diversity with the corresponding energy.
>
> ### Weakness 5 — Sequence-length scalability
>
> A paragraph at the end of Section 2.1 clarifies that the method imposes no length limit (one residue at a time conditioned on the prefix, so longer designs need only longer roll-outs and more data), and that we restrict to nanobody CDRs because the framework regions show little natural diversity. Extended discussion (full variable domains, paired chains, longer CDRH3) is in appendix A.13.
>
> ### Weakness 6 — Justification for the chosen entropy regularizer
>
> A paragraph at the end of Section 4.3 gives the intuition: the failure mode of Q-guided training is mode collapse onto a few high-Q sequences; maximizing a lower bound on the per-step marginal entropy of $\pi_\theta(a|s)$ directly penalizes this while staying agnostic to the discrete space and the oracle. We do not claim it is uniquely correct, but present it as one principled, application-agnostic baseline (a KL toward the natural CDR amino-acid distribution being a possible domain-specific alternative).
>
> ### Weakness 7 — Justification of the 2D VAE latent space
>
> Addressed by the same latent-dimensionality ablation (appendix A.11): sweeping $d \in \{1,\dots,5\}$ on Absolut! Expert, larger $d$ slightly improves affinity but lowers novelty and diversity (sharp novelty drop at $d \geq 3$), while $d=1$ underperforms. We adopt $d=2$ and note the alternative to a small continuous latent is not a discrete one (which would forfeit gradient-based Q-guidance). Section 4.1 acknowledges the bottleneck and points here.

---

### Review · Reviewer_5Dzq · 2026-04-07

**Summary Of Contributions:**

The paper studies offline antibody sequence design and proposes SeqDiff+QL, a sequential diffusion-policy method trained with a behavior-cloning term and Q-learning so that it can improve predicted affinity while staying close to the empirical training distribution.

The method uses a low-dimensional VAE latent space for amino acids, adds an entropy regularizer to improve the affinity-diversity tradeoff, and also studies two optional additions: contrastive biophysical priors in the latent space and post-generation filtering with learned Q-values. The experiments cover two in silico design tasks, based on Absolut! and Rosetta, multiple training distributions, five seeds, and several baselines, including BC, BC+QL, simultaneous diffusion, sequential diffusion, and GFlowNet

**Audience:**

Yes

**Audience Explanation:**

The paper sits at a useful intersection of offline reinforcement learning, diffusion-based generation, and biological sequence design. Even beyond the specific antibody application, the paper studies a question that is of general ML interest.

**Broader Impact Concerns:**

I do not see a major unaddressed ethical concern.

**Claims And Evidence:**

Yes

**Claims Explanation:**

The empirical section is fairly broad: the paper compares against several baselines, uses two design environments, evaluates different data regimes, runs five seeds, and includes ablations for entropy regularization, contrastive priors, and Q-value filtering. The main results are also nuanced rather than one-sided. For example, SeqDiff+QL is strongest on Absolut! expert and natural data and on Rosetta expert data, but it is not strongest on the random settings, where GFlowNet performs better. This makes the evidence more credible than a purely positive presentation.

At the same time, I do not think the current evidence fully supports the strongest wording in the abstract and conclusion. The paper supports improved predicted affinity under the selected evaluation oracles, not real binding performance. Also, the claim that the method “maintains diversity” is too strong without qualification: the main tables and figures show that Q-guided variants often lose diversity and novelty, and the entropy-regularized version only partly improves this tradeoff in selected settings. In addition, the Rosetta pipeline removes a small fraction of unstable evaluations, so the scope and limits of the evidence should be stated more carefully.

**Requested Changes:**

The paper should narrow its main claims in the abstract, introduction, and conclusion. The current experiments support improved predicted affinity under Absolut! and Rosetta, but they do not support claims about real experimental binding or therapeutic value. Likewise, the wording around “maintaining diversity” should be softened, because the main results show a clear drop in diversity and novelty under Q-guidance, with entropy regularization only partly recovering that tradeoff. (Critical)

The paper should support its central claim of constrained offline optimization more directly. At present, the argument that the policy stays near the data distribution is mainly based on the BC term and on indirect metrics such as novelty and duplicates. A more direct analysis of distribution shift or data proximity would make this claim much stronger. (Critical)

The Rosetta evaluation pipeline needs a clearer robustness discussion. The appendix states that a fraction of sequences yield unstable extreme values and are discarded. Please discuss how sensitive the conclusions are to this filtering choice, or provide a short sensitivity analysis if possible. (Critical)

Please add more analysis of the representation choices. The method depends on a two-dimensional amino-acid latent space and on a contrastive grouping prior. An ablation on latent dimensionality or simpler embeddings would help determine whether these choices are essential.

Please expand the analysis of Q-value filtering. This part is interesting, but it is only partly validated, and it clearly fails on random Rosetta data. Reporting Q-value calibration or ranking quality across all settings would make this component more convincing.

---

> ### Author Response · Authors · 2026-06-08
>
> We are grateful for the careful and balanced reading of our paper. All five requested changes are addressed; we summarize each below.
>
> ### (Critical) Narrowing of the main claims regarding affinity and diversity
>
> We agree the original wording overstated the in silico results and that "maintains diversity" was too strong. We narrow these claims throughout:
>
> - **Abstract.** Now frames affinity in terms of the Absolut! and Rosetta in silico oracles, and replaces the diversity wording with an explicit affinity–diversity tradeoff and the partial role of entropy regularization.
> - **Introduction.** The entropy-regularization contribution bullet now states the regularizer only *partially* mitigates the diversity loss from Q-guidance (Pareto-improved over part of the affinity range, with a residual cost at the highest affinities). A new closing paragraph in Section 1 states that all affinity claims refer to the Absolut!/Rosetta oracles, that no wet-lab validation is performed, and that results are not evidence of real binding or therapeutic value.
> - **Conclusion.** Section 6 now speaks of *oracle-predicted* affinity, acknowledges all evidence is computational, positions the method as a candidate generator that ranks well under specific oracles, and softens the diversity wording to partial mitigation.
>
> ### (Critical) Direct support for the constrained offline optimization claim
>
> New appendix subsection A.10 (Distribution-shift analysis) sweeps the Q-weight $\eta$ on Absolut! Expert and jointly reports six data-proximity indicators — Diversity, Novelty, mean Energy, Energy EMD to the training-set energy distribution, a Latent Distance (mean per-token minimum-L2 to the nearest training latent), and final-epoch BC loss — alongside position-wise logo plots at three values of $\eta$. As summarized at the end of Section 5.4.1, as $\eta$ grows the policy concentrates on high-Q latents, the Energy EMD first drops (entering the high-affinity tail of the training distribution) then rises (once it leaves support), and the BC loss grows — making the BC term's trust-region behavior explicit and quantitative.
>
> ### (Critical) Robustness of the Rosetta evaluation pipeline
>
> New appendix subsection A.9 (Sensitivity to the Rosetta energy threshold) recomputes every Rosetta table without the FastRelax outlier filter, for both antigens. Its first paragraph notes the filter shifts mean energies only marginally and never reverses a method comparison; we therefore keep it for noise reduction while reporting the unfiltered numbers for transparency. Forward references were added from Section 5.1 and from the appendix Rosetta task description.
>
> ### Latent-dimensionality ablation
>
> New appendix subsection A.11 (Latent dimensionality ablation) sweeps $d \in \{1,\dots,5\}$ on Absolut! Expert with all else fixed: $d \geq 3$ gives slightly better predicted affinity but a sharp novelty/diversity drop, while $d=1$ underperforms even after extended VAE training. We adopt $d=2$ and explain why a discrete representation is not the alternative (it would forfeit gradient-based Q-guidance). Section 4.1 now acknowledges the tight bottleneck and points here.
>
> ### Expanded analysis of Q-value filtering
>
> New appendix subsection A.12 (Q-value calibration) reports the Spearman correlation between learned Q-values and the oracle energy of *novel* sequences for SeqDiff+QL, BC+QL, and GFlowNet (mean $\pm$ std over seeds), for both antigens and all datasets. The paragraph at the end of Section 5.4.5 notes that Random Rosetta shows the lowest correlation, consistent with the lack of improvement from Q-value-based filtering there.

---

### Author Response · Authors · 2026-06-08
**Response to Reviewers**

We thank both reviewers for their careful and constructive reading of our manuscript. Their feedback substantially sharpened both the framing and the empirical scope of the paper. All changes made in response to this review round are highlighted in blue in the revised manuscript, so that revised text and newly added paragraphs, figures, and tables can be located at a glance.

To keep the main text concise and within scope, we have refrained from adding new figures to the main text; all new figures and tables are placed in the appendix and referenced from the corresponding main-text sections.

At a high level, the revision makes the following changes:

- **Narrowed all claims** to *oracle-predicted* affinity under the Absolut! and Rosetta in silico oracles, with no wet-lab validation, and softened the diversity claim to *partial* mitigation of the affinity–diversity tradeoff (abstract, introduction, conclusion).
- **Added a second antigen per oracle** (Absolut!: human CD38 / 3RAJ; Rosetta: PaaR2 N-terminal domain / 8C3K), with the same five seeds and identical hyperparameters, to probe cross-antigen generalization (new Section 5.4.3 + appendix).
- **Added a direct distribution-shift analysis** sweeping the Q-weight $\eta$ and reporting six data-proximity indicators alongside position-wise amino-acid logo plots (appendix; summarized in Section 5.4.1).
- **Added a Rosetta energy-threshold sensitivity analysis**, recomputing every Rosetta table without the FastRelax outlier filter (appendix).
- **Added a discussion of the GFlowNet inconsistency**, attributing it to proxy-oracle bias in the low-data regime (new Section 5.4.4).
- **Added a diversity-confound analysis**, quantifying the intrinsic affinity–diversity relationship on the 6.9M murine reference set, with an explicit caveat wherever diversity is reported (appendix + Section 5.4.1).
- **Added a latent-dimensionality ablation** ($d \in \{1,\dots,5\}$) justifying the 2D VAE bottleneck (appendix + Section 4.1).
- **Added a Q-value calibration analysis** (Spearman correlation across both antigens and all datasets) (appendix + Section 5.4.5).
- **Added intuition for the entropy regularizer** (Section 4.3) and a **CDR-length scalability discussion** (Section 2.1 + appendix).

Below we address each reviewer's concerns point by point, indicating where in the revised manuscript the changes can be found. We refer to revised sections by title and number.